## TECHNIQUE

# Validation of aerial photogrammetry methods to measure body size, condition and mass in small cetaceans

Riccardo Cicciarella[1] , Erik P. Willems[1] , Benjamin Markham[2] , Manuela R. Bizzozzero[1] ,
Wayne Phillips[2] , Simon J. Allen[1,3,4] , Michael Krützen[1] and Fredrik Christiansen[5]

[1]*Evolutionary Genetics Group, Department of Evolutionary Anthropology, University of Zurich, Zurich, Switzerland*
[2]*Marine Sciences, Sea World Foundation Australia, Main Beach, QLD, Australia*
[3]*School of Biological Sciences, University of Bristol, Bristol, UK*
[4]*School of Biological Sciences, Oceans Institute, University of Western Australia, Crawley, WA, Australia*
[5]*Marine Mammal Research, Department of Ecoscience, Aarhus University, Roskilde, Denmark*

Handling Editors: Eleonora Grandi & Janna Morrison

The peer review history is available in the Supporting Information section of this article (https://doi.org/10.1113/JP290419#support-information-section).

**Abstract figure legend** Unoccupied aerial vehicle (UAV)-based photogrammetry provides measurements equivalent to traditional hand measurements in bottlenose dolphins. Mass estimates derived from UAV measurements closely match the true body mass of live individuals.

M. Krützen and F. Christiansen contributed equally to this work.

**Abstract** Accurate morphometric measurements are essential for estimating body size and condition in animals. These characteristics are, in turn, key to eco-physiological studies, wildlife management and conservation. For free-ranging cetaceans, however, collecting non-invasive morphometric data is challenging. Unoccupied aerial vehicle (UAV) photogrammetry offers a promising solution but requires ground-truthing to assess accuracy and precision. Similarly, morphometric-based indices of body condition must be validated against the animals' true body condition. Here we validated UAV-derived estimates of body size and condition in bottlenose dolphins (*Tursiops* spp.) under human care by comparing photogrammetry-based measurements of body length, width, height and girth from both stationary and swimming individuals with manual measurements. The two methods showed negligible differences, with UAV-based data yielding lower variability, confirming both high measurement accuracy and precision. Using UAV-derived measurements we calculated a volume-based body condition index (BCI) and compared it with a mass-based BCI, a standard metric in ecological research. The two indices showed a near-perfect fit, demonstrating that volume-based metrics reliably reflect true body condition in small cetaceans. Body density decreased with increasing body condition, consistent with higher fat-to-muscle ratios. By combining UAV-derived body volume with predicted density, based on their body condition, we accurately estimated individual body mass (mean error = 6.4%). This study provides a comprehensive validation of UAV-based photogrammetry to estimate body size, condition and mass in small cetaceans, highlighting its value as a non-invasive and cost-effective tool for ecological and conservation research.

(Received 30 October 2025; accepted after revision 7 January 2026; first published online 30 January 2026)

**Corresponding author** R. Cicciarella and Michael Krützen: Evolutionary Genetics Group, Department of Evolutionary Anthropology, University of Zurich, 8057 Zurich, Switzerland. Email: riccardo.cicciarella@iea.uzh.ch and michael.kruetzen@iea.uzh.ch

## Key points

- Measuring body size and condition in free-ranging dolphins is difficult, yet essential to understand their physiology, energy reserves and health.
- We used unoccupied aerial vehicles (UAV) to obtain accurate, non-invasive body measurements of bottlenose dolphins and compared them with direct manual measurements.
- UAV-based photogrammetry produced highly precise and accurate estimates of body length, girth and overall body volume, even for freely swimming animals.
- A UAV-derived, volume-based body condition index matched traditional mass-based indices and enabled accurate estimation of body mass.
- These results validate UAV photogrammetry as a reliable, ethical and cost-effective method for assessing body size, condition and mass in small cetaceans, thereby advancing ecological and physiological research in the wild.

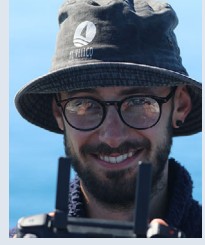

After completing his graduate studies at the University of Massachusetts Dartmouth, **Riccardo Cicciarella** began his PhD in the Department of Evolutionary Anthropology at the University of Zurich, where he investigates the bioenergetics of Indo-Pacific bottlenose dolphins in Shark Bay, Western Australia. His research integrates UAV photogrammetry, physiological modelling and long-term behavioural data to assess how energetic constraints shape mating strategies and individual fitness. Riccardo aims to advance non-invasive methods for evaluating wildlife health and population resilience, with the broader goal of improving conservation strategies in rapidly changing ecosystems. He is committed to science communication to make research accessible beyond academia.

## Introduction

Body size and condition are two key morphological traits that together characterize an organism's physical structure and physiological state. Body size refers to the structural dimensions of an individual, typically measured as body length or mass, and is a primary determinant of metabolic rate and life history traits (Bernstein, 2010; Chown & Gaston, 2010). Body condition, in contrast, reflects an individual's energy reserves relative to its structural size (Hanks, 1981; Millar & Hickling, 1990), serving as an indicator of nutritional state and physiological fitness. Both traits are critical for understanding animal ecology, physiology and responses to stressors (Dobson, 1992; Garland, 1983; Lauder, 1981; Stevenson & Woods, 2006).

Metabolic rates in animals scale allometrically with body size, which in turn affects food requirements, growth, reproduction and mortality (Anderson-Teixeira et al., 2009; Kleiber, 1947). Body size also governs physiological functions such as heat loss (Phillips & Heath, 1995; Porter & Kearney, 2009), fasting endurance (Lindstedt & Boyce, 1985) and the energetic costs of locomotion (Garland, 1983; Irschick & Higham, 2016).

Accurate measurements of body size are therefore fundamental to understanding the biology and ecology of wild populations. Body length (BL), in particular, serves as a reliable predictor of age class (Vivier et al., 2023), enabling researchers to classify individuals into age groups and assess key demographic parameters such as survival and reproductive rates (Holmes & York, 2003). BL can also serve as a proxy for growth rate in many vertebrates (Andrialovanirina et al., 2020; Davis et al., 2008; Mott et al., 2010). More broadly, body size reflects ecological processes and population status and is frequently used as a basis for assessment and management efforts.

Body condition provides insights into the energetic demands of life functions including body maintenance, growth, mating and reproduction (Anderson-Teixeira et al., 2009; Kleiber, 1947). In females, body condition is often positively related to reproductive success, including birth size, offspring condition, growth rate and survival (Christiansen et al., 2018; Taillon et al., 2012; Tauson, 1993). Body condition can also serve as an indicator of the energetic costs of gestation and lactation. In Southern right whales (*Eubalaena australis*), for example, lactating females lose about 25% of their body volume over a 3-month breeding season, with body condition loss being positively related to calf growth rate (Christiansen et al., 2018).

In males, better body condition can enhance reproductive success by enabling individuals to sustain energetically costly mating behaviours for longer periods. This has been observed across taxa, including bullfrogs (*Rana catesbeiana*; Judge & Brooks 2001), grey seals (*Halichoerus grypus*; Lidgard et al. 2005) and rhesus macaques (*Macaca mulatta;* Higham et al. 2011). In these species, males in superior condition show greater endurance in mate competition and achieve higher mating success. At the population level, body condition serves as a key indicator of nutritional health and can reflect shifts in vital rates and demographic trends in response to environmental or anthropogenic pressures (Dobson, 1992; Rolland et al., 2016). Assessing body condition at both the individual and population levels is therefore essential for wildlife monitoring and conservation (Stevenson & Woods, 2006).

In cetaceans, body size and condition are commonly estimated by taking morphometric measurements, collected during whaling operations or from stranded or temporarily restrained individuals (Barratclough et al., 2019; Fortune et al., 2012; Lockyer, 1981). These include BL (Gómez-Campos et al., 2011; Lockyer & Waters, 1986), which serves as a proxy for structural size. Body condition can be assessed using methods that directly estimate energy reserves, including blubber thickness (Cartee et al., 1995), blubber composition (Aguilar & Borrell, 1990) or relative adipocyte volume (Castrillon et al., 2017), as well as biochemical techniques such as carcass analysis (Iverson et al., 2010) and isotope dilution (Bowen & Iverson, 1998). Although these methods have provided valuable insights and are used in health assessment programmes for population-level monitoring (Wells et al., 2004), they rely on captured or stranded animals and therefore preclude long-term monitoring of small, remote or endangered populations. They may also introduce sampling biases towards specific age, reproductive classes or body condition (Aguilar et al., 1999; Krahn et al., 2004), and post-mortem distortion in stranded animals compromises carcass-based measurements (Barratclough et al., 2014; Moore et al., 2004). These limitations underscore the need for accurate, non-invasive and non-lethal methods to assess morphology and condition in wild marine mammals.

Measuring body size and condition in wild cetaceans presents major logistical challenges due to their aquatic lifestyle (Ball et al., 2017; Iverson et al., 2010). Although smaller cetaceans can be captured or restrained, these procedures are logistically complex and induce significant stress (Rojas-Bracho et al., 2019). In many regions, however, capture or restraining is not feasible even for small species due to factors such as water depth or turbidity, unsafe handling conditions, high financial and personnel requirements or ethical constraints associated with increased risk to both animals and researchers (Norman et al., 2004). The challenges are amplified for species that are elusive, inhabit remote areas or endangered. For larger marine mammals the risks associated with close contact often preclude even temporary restraint (Hunt et al., 2013).

To address these challenges researchers have developed novel methods to estimate body size and condition (Castrillon & Bengtson Nash, 2020). These include metrics based on relative adipocyte volume (Castrillon et al., 2017; Druskat et al., 2019) and measurements of blubber thickness using ultrasound technology, a method successfully applied to free-swimming belugas (*Delphinapterus leucas*) and right whales (Cornick et al., 2016; Miller et al., 2011).

In addition, unoccupied aerial vehicle (UAV)-based photogrammetry is increasingly used as a non-invasive approach to measure body morphometrics in living marine mammals. This method has already been applied to large cetaceans (Christiansen et al., 2019; Glarou et al., 2023), smaller cetaceans (de Oliveira et al., 2023), pinnipeds (Hodgson et al., 2020) and sirenians (Ramos et al., 2022), demonstrating its utility to non-invasively monitor body size and condition, seasonal variations, reproductive costs and health status. However, despite its widespread use, rigorous validation of UAV-derived morphometrics against direct measurements from living animals is still lacking. Ideally, such validation involves comparing UAV-based measurements with concurrent manual measurements taken on the same individuals. Similarly, body condition metrics based on UAV-derived morphometrics need to be validated by comparing them to standard mass-based metrics, which are commonly used in ecophysiological studies (Green, 2001; Peig & Green, 2009; Schulte-Hostedde et al., 2005). Although Zhang et al. (2023) demonstrated that 3D and elliptical models can accurately estimate the body condition and mass of dead porpoises, further validation using live, free-swimming animals is essential to apply UAV-based methods for monitoring health, reproduction and survival in wild populations.

This study aimed to validate UAV-based photogrammetry for measuring body morphology, assessing body condition and estimating body mass (BM) in two small cetacean species (*Tursiops* spp.) under human care. Specifically, we (1) compared UAV-derived morphometric measurements with manual measurements, (2) validated volume-based body condition metrics from UAV photogrammetry against mass-based body condition indices and (3) evaluated the accuracy of UAV-derived mass estimates for stationary and free-swimming dolphins.

## Materials and methods

### Ethical approval

All work involving dolphins at Sea World took place within the facility's established animal care, husbandry and veterinary programmes. Routine morphometric measurements were provided by Sea World staff as part of their ongoing health practices and were obtained under the facility's Exhibition Animals Permit (PRID000821) and its Research Permit authorizing scientific use of protected wildlife (WA0035654), issued by the Queensland Department of Environment and Science. Data were collected in coordination with Sea World under its permitted research activities and in accordance with its operational guidelines for research on animals. No procedures required handling beyond standard husbandry.

### Data collection and morphometric measurements

Morphometric data were collected on 14 *Tursiops truncatus* ($n = 8$ females and $n = 6$ males) and 11 *T. aduncus* ($n = 5$ females and $n = 6$ males) under human care at the Sea World in Gold Coast, Queensland, Australia. The dolphins were housed in lagoons with sandy (natural) substrate and supplied with water sourced from natural waterways. The dolphins were maintained in both same- and mixed-sex groups. Dolphins were fed a seasonally adjusted, individually tailored diet consisting of a minimum of five different fish species daily, all sourced through human-grade fish brokers. Rations varied based on individual energetic requirements and ranged approximately from 5.5 to 15 kg of fish per animal per day. Commonly fed fish species are submitted for nutritional analysis twice annually to ensure accurate characterization of the diet's nutritional composition.

We collected aerial photogrammetry measurements in 2022 using an Inspire 2 multirotor UAV (DJI, Shenzhen, China), equipped with a Zenmuse X5S CMOS four-thirds camera (20.8 effective megapixels; DJI) with a 25 mm f/1.8 M.Zuiko lens (Olympus, Tokyo, Japan). In 2023 we used a Mavic 3 multirotor UAV (DJI), equipped with a CMOS four-thirds camera (20 effective megapixels; Hasselblad, Gothenburg, Sweden) with a 24 mm f/2.8 to f/11 lens. We operated UAVs in 2022 and 2023 at altitudes of 25–35 m and 15–25 m, respectively. Both UAV were equipped with a SF11/C laser range finder (LightWare LiDAR, Centurion, South Africa) to measure the altitude above the focal animals.

When recording the dolphins, we angled the UAV cameras vertically at −90° (zenithal angle). The video resolution was 4K@60fps for the Inspire 2 and 5.1K@50fps for the Mavic 3. We recorded both the dorsal and lateral sides of the dolphins when they were held stationary by the trainers assisting the dolphins in rotating their bodies in the water to either the left or the right side in the shallow-water area of the pool.

We also obtained video recordings while the animals were swimming freely near the surface to simulate conditions of monitoring wild dolphins. From the dorsal images, we measured BL (distance from tip of rostrum to the end of the tail notch) and widths (W, at 5% increments

along the body axis; Christiansen et al. 2016). From the lateral images we measured body height (H, dorso-ventral distance) at the same measurement sites as the width measurements (Christiansen et al., 2019). Per animal, we extracted a total of 19 width and height measurements from the imagery, using MorphoMetrix (Torres & Bierlich, 2020).

Trainers at Sea World obtained manual measurements using a measuring tape. Measurements included BL and five straight-line measurements of body width and height at five defined body positions (AP = immediately anterior to pectoral fin, AD = immediately anterior to dorsal fin, PD = immediately posterior to dorsal fin, MG = mid-genital and MP = mid-peduncle, similar to Noren et al. 2019; Fig. S1). The trainers took measurements on the dorsal and on one lateral side of each animal. They also measured body girth at the same five body positions. Each trainer measured each individual dolphin once, resulting in three repeated measurement per dolphin.

For the UAV measurements, we selected still frames from the UAV video recordings and used MorphoMetriX to place reference points on the dorsal and one lateral side at the same five body positions. The software calculated distances between reference points in pixels and converted them to real-world measurements using camera metadata and the ground sampling distance (Torres & Bierlich, 2020). We used images of UAV video recordings when the dolphins were held stationary and when the dolphins were swimming freely near the surface. Three different trainers each measured every individual once, resulting in three repeated measurements per dolphin. The dorsal (width) and lateral (height) side measurements were then used to estimate the girth of each animal at the five body positions using the formula by Byrd and Friedman (2013):

$$G_i = 4 * \int_0^{\frac{\pi}{2}} \text{sqrt}\left(r_{W,1}^2 * \cos(x)^2 + r_{H,i}^2 * \sin(x)^2\right) dx \quad (1)$$

where $r_W$ represent the radius of the body width ($W$) and $r_H$ represent the radius of the body height ($H$) of the dolphin $i$ at the position of the girth measurement (i.e. AP, AD, PD, MG and MP).

The trainers recorded the BM of each dolphin by training them to slide onto a shallow weigh station. The scale was then raised out of the water and, when the water drained off, the weight was recorded. All animals were weighted within a maximum of 10 days after the collection of the UAV data.

## Body volume

To estimate body volume (BV), we first calculated the body height-to-width (HW) ratio at each 5% measurement site for each dolphin (Christiansen et al., 2019). All analyses were conducted in R (R Core Team,

2020). We used a linear mixed-effects model using the package 'glmmTMB' (Brooks et al., 2017) to test whether the HW ratio was affected by species, BL (adjusted by subtracting the species-specific mean from each individual according to species to address multicollinearity), body condition (expressed as the relative body widths W at 40% of BL measured from the rostrum) or sex with measurement site and individual ID included as random effects. We used body width at 40% BL as a proxy for body condition because this location showed the greatest variation in body width among the dolphins in our study. We then estimated the BV of each dolphin from their BL, widths (W) and heights (H) measurements, using the segmented elliptical model of Christiansen et al. (2019):

$$\text{BL}_i * 0.05 * \int_0^1 \pi * \frac{W_{A,s,i} + \left(W_{P,s,i} - W_{A,s,i}\right) * x}{2}$$

$$* \frac{H_{A,s,i} + \left(H_{P,s,i} - H_{A,s,i}\right)}{2} dx, \quad (2)$$

$$\text{BV}_{\text{obs},i} = \sum_{s=1}^{20} V_{s,i}, \quad (3)$$

where $V_{s,i}$ represents the volume of the segment $s$ ($S = 20$ in total) for individual $i$, $\text{BL}_i$ is the total BL, $W_{A,s,i}$ and $W_{P,s,i}$ are the anterior and posterior $W$ measurements, respectively, and $H_{A,s,i}$ and $H_{P,s,i}$ are the anterior and posterior $H$ measurements, respectively. The equation inside the integral represents the area of an ellipse ($A = \pi * r_1 * r_2$), where $r_1$ is the major radius (given by the first quotient) and $r_2$ is the minor radius (given by the second quotient; Christiansen et al., 2019).

To represent the gradual decrease in W and H towards the tail region of the dolphin and the rostrum area, we assigned a W and H of 0 m to the end points of the dolphins (0 and 100% BL from the rostrum). We replaced the W and H measurements at 90 and 95% BL from the rostrum with linearly interpolated values between the 85 and 100% BL measurement sites (Christiansen et al., 2019). The third formula expresses the observed (measured) body volume $\text{BV}_{\text{obs}, i}$ of the individual $i$ by summing the 20 volume segments $V_{s,i}$ calculated in Eqn 2.

Stationary dolphins were measured under optimal conditions, whereas images of freely swimming dolphins may introduce errors from water distortion and splashes, resulting in less precise BL and W measurements. Additionally, variations in body posture, such as degree of straightness of body along its horizontal axis, body roll, arch and pitch, can also affect the measurement accuracy, as previously shown (Christiansen et al., 2018). To assess whether these factors influence the calculation of BV, we performed a linear model using the package 'lmer' (Bates

et al., 2015), comparing the volumes derived from W measurements of both stationery and swimming dolphins.

## Volume-based body condition

We calculated body condition index (BCI) following the methods of Christiansen et al. (2018):

$$BCI_i = \frac{BV_{obs,i} - BV_{exp,i}}{BV_{exp,i}}, \qquad (4)$$

where $BV_{obs,i}$ is the observed (measured) BV of the dolphin $i$ (in m$^3$) and $BV_{exp,i}$ is the predicted (average) BV of the dolphin $i$ (in m$^3$) derived from the linear relationship between BV and length after a log-log transformation.

The BCI accounts for the structural size (Peig & Green, 2009), for example, BL of the dolphins and, hence, represents the relative energy reserves of an individual irrespective of its BL. To demonstrate this, we also calculated a standardized $BCI_{std}$ based on the BV estimated when the BL was fixed to 1 and using the relative body widths (W) and heights (H), both expressed as a proportion of BL, instead of absolute values (Christiansen, Dawson et al., 2020):

$$BCI_{std} = \frac{BV_{std,i} - BV_{std\,\mu}}{BV_{std\,\mu}}, \qquad (5)$$

where $BV_{std,i}$ is the standardized BV of the dolphin $i$ (in m$^3$) and $BV_{std\,\mu}$ is the average of the standardized BVs of all individuals.

We used a linear model with the package 'lmer' (Bates et al., 2015) to compare the BCI and $BCI_{std}$ estimates. A positive linear relationship with an intercept of 0 and a slope parameter of 1 would indicate a perfect fit between the two indices. Finally, to assess whether measurements from stationary and freely swimming dolphins affect not only BV but also BCI, we compared BCI derived from images of stationary and freely swimming dolphins. Given the potential measurement variation discussed previously (Body Volume section) regarding BV, we tested whether these same factors influence the accuracy of BCI estimates.

## Body mass and density

BM plays a crucial role in bioenergetic models, as it improves the estimation of body condition when considered alongside structural size (Dobson & Michener, 1995; Dobson et al., 1999). For each dolphin, we obtained BM by weighing individuals on a platform scale. We then calculated body density (BD) by dividing the measured BM by the estimated BV of the same individual. Next we investigated the relationship between BD and BCI using linear models from the package 'lmer' (Bates et al., 2015). As cetaceans deposit most of their energy

reserves as blubber, which is less dense than muscle, we expected a negative relationship between BD and BCI (Aoki et al., 2021; Castrillon & Bengtson Nash, 2020). This relationship is represented by the following formula:

$$BD_i = \alpha - \beta * BCI_i, \qquad (6a)$$

where $\alpha$ and $\beta$ represents the intercept and slope parameters of the linear model, respectively. Finally, to predict body mass ($BM_{pred}$) without requiring direct weighing, we multiplied the observed body volume ($BV_{obs}$) by BD. By combining Eqns 3, 4 and 6, we estimated $BM_{pred}$ as follows:

$$BM_{pred} = BV_{obs} * BD \qquad (7a)$$

$$BM_{pred} = BV_{obs} * \left( \alpha - \beta * \frac{BV_{obs,i} - BV_{exp,i}}{BV_{exp,i}}, \right). \quad (7b)$$

## Validation of morphometric measurements

**Body lengths, widths, girths and inter-observer variability.** We used Krippendorff's Alpha ($\alpha_{Krip}$) to assess inter-observer reliability for measuring BL, W and girth at the five body sites AP, AD, PD, MG, MP, as implemented in the package 'krippendorffsalpha' (Hughes, 2022). We conducted this analysis for (1) hand measurements by trainers, (2) UAV measurements of stationary animals and (3) UAV measurements of swimming animals. For body girth measurements, we assessed inter-observer reliability only for hand measurements by trainers, and we compared trainer measurements with UAV measurement of stationary animals. Additionally, we used Krippendorff's Alpha to evaluate agreement across the three measurement methods ('Trainers', 'UAV Stationary' and 'UAV Swimming') for BL, W and girths.

To evaluate the accuracy of UAV measurements compared to hand measurements, we used a generalized linear mixed-effects model (GLMM) fitted with the 'glmmTMB' package (Brooks et al., 2017) to simultaneously express the mean and variance of BL and W measurements at the five body positions (AP, AD, PD, MG and MP). The dependent variable was morphometric measurement, and the fixed effects included measurements methods (categorized as 'Trainers', 'UAV Stationary' and 'UAV Swimming'), UAV type (Inspire 2 or Mavic 3), species (*T. truncatus* or *T. aduncus*), body position (BL, AP, AD, PD, MG, MP) and the interaction between species and site. We accounted for individual ID and observer variability by including them as random factors. To address rank deficiency due to overlapping UAV and method categories (the method 'Trainer' necessarily lacked a value for UAV type, thus creating multicollinearity), we created a composite

variable (Method-UAV), combining method and UAV type. This allowed for contrasts to compare measurement methods and UAV types across different contexts. We used an ANOVA to evaluate the significance of fixed effects in the package 'stats' (R Core Team, 2020).

**Comparing mass- and volume-based body condition.** To test the performance of our volume-based BCI, a comparison with the mass-based BCI of the dolphins was necessary. Therefore, we tested our volume-based BCI against the more commonly used mass-based BCI. To achieve that, we first estimated a BCI based on body mass ($BCI_{mass}$):

$$BCI_{mass} = \frac{BM_{obs,i} - BM_{exp,i}}{BM_{exp,i}}, \tag{8}$$

where $BM_{obs,i}$ is the observed (measured) BM of the dolphin $i$ (in kg) and $BM_{exp,i}$ is the predicted (average) BM of the dolphin $i$ (in kg) derived from the linear relationship between BM and length after a log-log transformation.

We compared our volume-based BCI (Eqn. 4) to the mass-based BCI (Eqn. 8) using linear models with the package 'lmer' (Bates et al., 2015). An intercept near 0 and a slope close to 1 would indicate a perfect fit between the two metrics.

**Assessing the accuracy of the body mass model.** To assess the accuracy of our BM model for dolphins, we compared the estimated BM of dolphins from Eqn. 7 with the direct measurement of BM recorded by the trainers under two scenarios: (1) while stationary and (2) while swimming freely near the surface (to simulate typical situations for wild dolphins). To estimate the dolphins' body volume (BV), we used the measured BL and W at the 19 measurement sites from UAV footage, while H was predicted from the corresponding W using the mean height:weight (HW) ratio for each measurement site. We used predicted H over measured H because the latter is rarely obtainable in wild cetaceans and must typically be estimated from the corresponding W measurement (Christiansen et al., 2019). From the resulting BV estimates, we estimated BCI and BM for both stationary and swimming photographs. Finally, we used two linear models using the package 'lmer' (Bates et al., 2015) to compare true BM with the BM estimated from W measurements taken stationarily and while swimming.

## Results

### Dolphin body morphometric measurements

The body shape of the measured dolphins seems to be vertically elongated (Fig. 1). The W of the animals showed a steep increase in the head region, rising rapidly from 5% to 25% of the BL, and reaching its maximum between 30% and 35% BL. Beyond this point, W gradually decreased with a gentler slope from 40% to 85% BL (Fig. 1*A*). In contrast, H increased steeply in the head region from 5% to 25% BL, followed by a more moderate increase and decrease between 30% and 55% BL, peaking at 45% BL. After 55% BL, H decreased sharply between 60% and 70% BL, before tapering off again in the tail region (Fig. 1*B*). It appeared that variation in W and H among individuals was generally larger at between 30 % and 60% of BL (Fig. 1*A* and *B*). The body height:weight (HW) ratio flattened over the lateral plane of the dolphins' body axis (Fig. 1*C*).

The body shape was rounder from the rostrum to approximately 35% BL, with the mean HW ratios ranging from 1.11 to 1.32. (Fig. 1*C*; Table S1). The lateral flattening increased from 40% to 70% BL, with mean HW ratios ranging from 1.31 to 1.66, and became more extreme in the peduncle region from 75% to 85% BL, with ratios between 1.91 to 3.5 (Fig. 1; Table S1). The HW ratios were unaffected by BL, BCI (measured as W at 40% BL) and did not differ between species or sexes (Table S2).

### Body volume

BV estimates of bottlenose dolphins ranged from 0.105 to 0.155 m$^3$ in *T. aduncus* and from 0.168 to 0.297 m$^3$ in *T. truncatus* (Table S3). There was a strong positive relationship between UAV-derived BV measurements of the stationary versus freely swimming dolphins ($F_{(1, 23)} = 262$, $P < 0.001$, $R^2 = 0.92$; Fig. 2), suggesting that BV can be measured reliably in either condition. The intercept of the relationship was $-0.0095 \pm 0.012$, and the slope parameter was $1.02 \pm 0.063$. These results indicate that BV estimates are consistent, regardless of whether the dolphins are stationary or swimming.

### Volume-based body condition

There was a strong linear relationship between estimated BV and BL on the log-log scale ($F_{(1, 23)} = 157.1$, $P < 0.001$, $R^2 = 0.87$; Fig. 3*A* and *B*) indicating that BV increases with length following an exponential trajectory:

$$\log\left(BV_{exp,i}\right) = -4.02 + 2.59 * \log\left(BL_i\right) \tag{9}$$

The BCI of the dolphins in this study varied from $-0.186$ to $0.267$ (Table S3). The BCI values based on absolute size (Eqn. 4) and the standardized index based on relative size ($BCI_{std}$, Eqn. 5) were highly correlated ($F_{(1, 23)} = 143.7$, $P < 0.001$, $R^2 = 0.86$; Fig. S2), with an intercept of $0.0057 \pm 0.0084$ and a slope parameter of $0.87 \pm 0.073$. These results suggest that BCI accounts for the structural dimensions of an individual independently of its BL and, therefore, provides an unbiased estimate of its

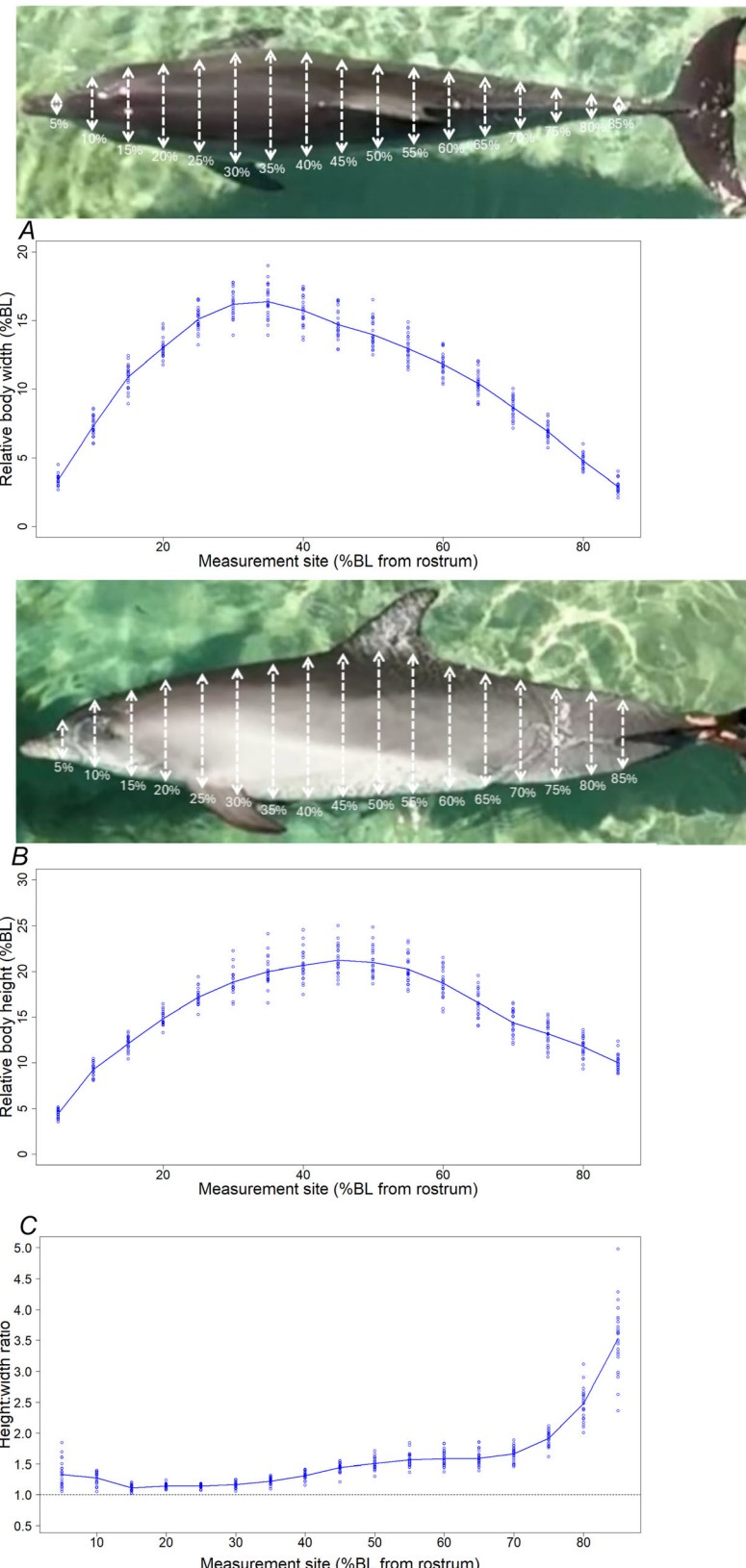

**Figure 1. Cross sectional body shape of bottlenose dolphins (*Tursiops* spp.)**
*A*, body width (W) measured from 5% to 85% of overall body length (BL). *B*, body height (H) measured from 5% to 85% of overall BL. *C*, body height:weight (HW) ratio from 5% to 85% of overall BL. The dashed black line shows a 1:1 ratio which represents a circular cross-sectional body shape. Solid blue lines indicate the average value among all animals, whereas blue circles represent each individual measurement (*N* = 25 individuals).

**Table 1. Krippendorff's alpha estimates for inter-observer reliability and method comparison for body length (BL), body width (W) and girth measurements.**

|  | Body length and width | | | | Body girth | | | |
|---|---|---|---|---|---|---|---|---|
|  | Estimate | Lower 95%CI | Upper 95%CI | *n* | Estimate | Lower95%CI | Upper 95%CI | *n* |
| Hand measure by trainers | 0.973 | 0.963 | 0.981 | 445 | 0.974 | 0.952 | 0.982 | 350 |
| UAV measure of stationary animals | 0.981 | 0.974 | 0.987 | 450 | – | – | – | |
| UAV measure of swimming animals | 0.986 | 0.981 | 0.99 | 450 | – | – | – | |
| Methods comparison | 0.973 | 0.969 | 0.977 | 1335 | 0.945 | 0.928 | 0.96 | 375 |

Abbreviation: UAV, unoccupied aerial vehicle.

relative energy reserves. Additionally, BCI calculated from morphometric measurements (BL, W and H predicted from height:width ratio) of stationary and swimming dolphins showed a significant positive correlation ($F_{(1, 23)}$ = 28, $P < 0.001$, $R^2 = 0.53$, intercept = 0.00065 ± 0.013, slope = 0.65 ± 0.12; Fig. 3*F*). These results suggest that BCI estimates were consistent, regardless of whether the dolphins were stationary or swimming freely, highlighting its potential in the wild.

## Body mass and density

The true BM of the sampled bottlenose dolphins, measured manually, ranged from 113 to 283 kg (Table S3). The estimated BD of bottlenose dolphins ranged from 904

to 1116 kg m$^{-3}$ (Table S3). There was a significant negative relationship between BD and BCI ($F_{(1, 23)}$ = 5.1, $P = 0.03$, $R^2 = 0.15$; Fig. 4), suggesting that animals with higher body condition values have lower BD. The slope and intercept of this relationship were substituted from Eqn. 6a to obtain the following equation:

$$BD = 1000.71 - 278.12*BCI \qquad (6b)$$

## Validation of morphometric measurements

**Inter-observer variability.** Inter-observer reliability values for BL and W measurements were very high for both the UAV-based and manual approaches (Table 1). Within methods, hand measurements by trainers

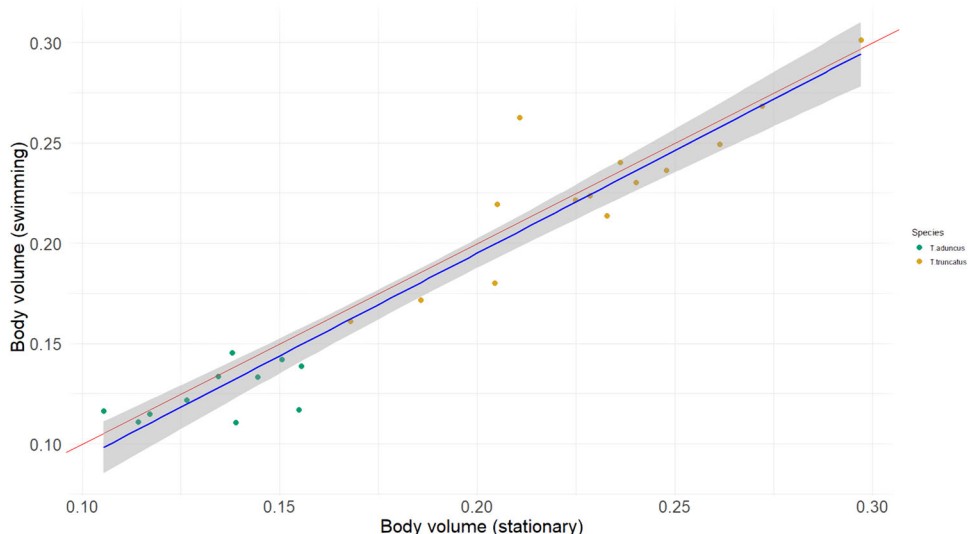

**Figure 2. Linear relationship between estimates of body volume from stationary and freely swimming bottlenose dolphins**
The red line represents a 1:1 relationship, whereas the blue line represents the fitted values of the linear model (intercept = −0.0095 ± 0.012, slope = 1.02 ± 0.063) with the shaded area showing the 95% confidence interval. *N* = 25.

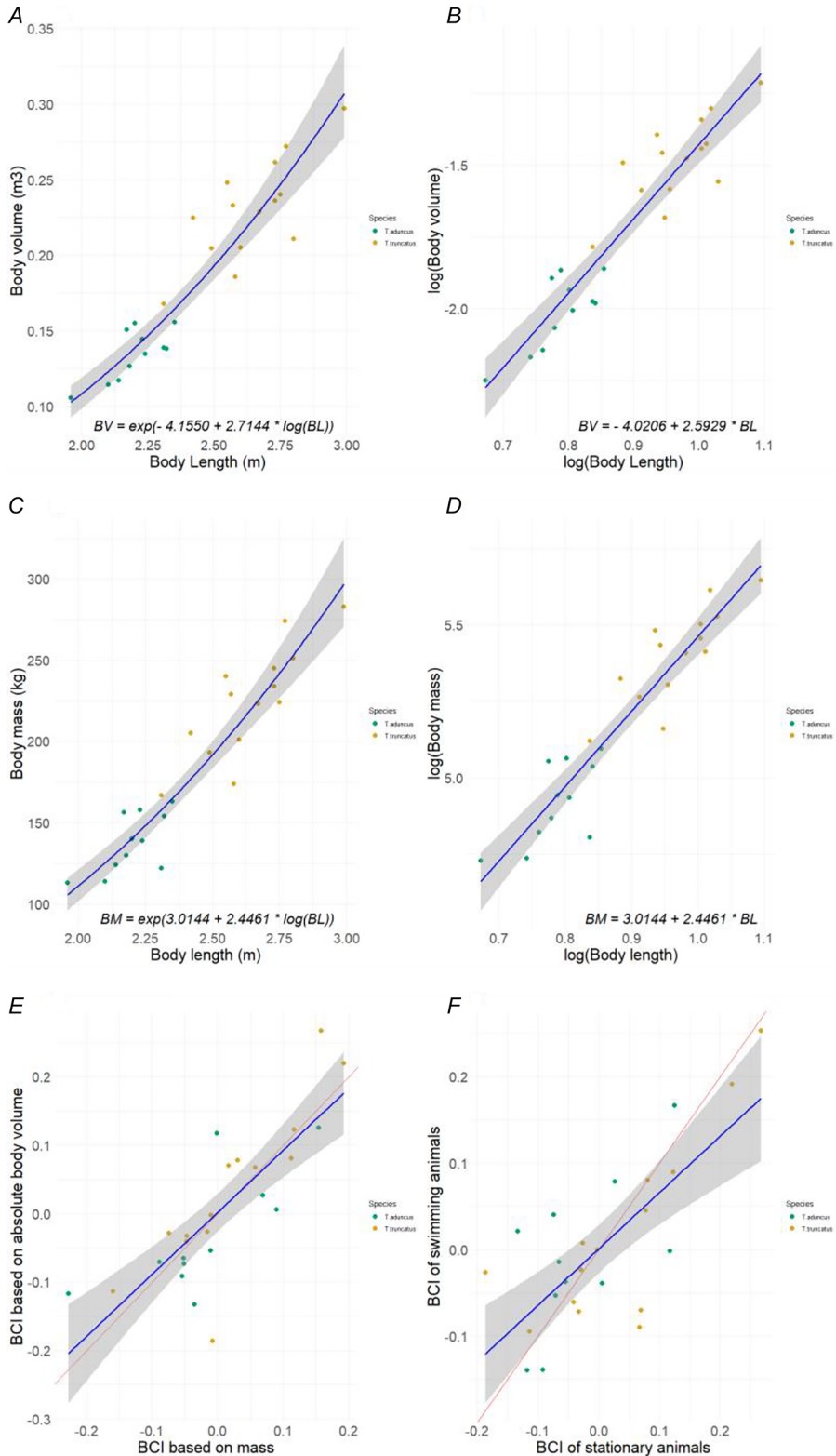

**Figure 3. Relationships among body size, body mass and body condition indices in bottlenose dolphins**
*A*, body volume (BV) as a function of body length (BL). *B*, log–log relationship between BV and BL. *C*, body mass (BM) as a function of BL. *D*, log–log relationship between BM and BL. *E*, linear relationship between volume-based

body condition index (BCI$_{BV}$) and mass-based BCI$_{BM}$ (intercept = 0.0014 ± 0.013, slope = 0.91 ± 0.014). *F*, linear relationship between BCI estimated from unoccupied aerial vehicle (UAV) measurements of stationary and freely swimming bottlenose dolphins (intercept = 0.00065 ± 0.013, slope = 0.65 ± 0.12). The red lines represent a 1:1 relationship, whereas the blue line represents the fitted values of the linear models with the shaded area showing the 95% confidence interval. *N* = 25.

showed strong agreement ($\alpha_{Krip}$ = 0.973), as did UAV measurements of stationary ($\alpha_{Krip}$ = 0.981) and freely swimming dolphins ($\alpha_{Krip}$ = 0.986). Between methods, all three approaches ('Trainers', 'UAV Stationary' and 'UAV Swimming') also demonstrated high agreement for BL and W measurements ($\alpha_{Krip}$ = 0.976), highlighting their equivalence. Similarly, girth measurements taken by trainers exhibited high inter-observer reliability ($\alpha_{Krip}$ = 0.974). The comparison of girth measurements across methods yielded slightly lower, yet still strong agreements ($\alpha_{Krip}$ = 0.945).

A location-dispersion GLMM revealed significant differences between species in body measurements in terms of both their mean value (location; Fig. 5*A*), and variance around these mean values (dispersion; Fig. 5*B*). Mean BL and W measurements for *T. truncatus* were consistently larger than those for *T. aduncus* (Fig. 5*A*; Table S4), whereas measurements at the different body locations also differed (Chi$^2$df = 536.62, *P* < 0.001; Table 2). No significant differences were attributed to the measurement method, with the exception for the Mavic 3, for which estimates obtained in the stationary condition were marginally (4 mm) but significantly higher than in the swimming condition (Table S4). Although statistically detectable, this difference is unlikely to be

**Table 2. ANOVA Chi-Square test for the generalized linear mix model examining the effects of UAV, species, body position and the interaction between species and body position on bottlenose dolphin body length (BL), and width (W) measurements.**

| Predictor | Chi-Square | Df | *P*-value |
|---|---|---|---|
| UAV | 39.743 | 4 | <0.001 |
| Species | 43.116 | 1 | <0.001 |
| Body position | 81084.885 | 5 | <0.001 |
| Species:body position | 536.621 | 5 | <0.001 |

Abbreviation: UAV, unoccupied aerial vehicle.

biologically meaningful (Fig. 5*B*; Table S4). Accuracy of UAV measurements was thus comparable to that of hand measurements.

Variation in body measurements was lower for UAV-based methods than for hand measurements, particularly when using a Mavic 3 (Fig. 5*B*). The UAV-based methods hence appear to have higher precision than hand measurements.

**Comparing mass- and volume-based body condition.** There was also a strong linear relationship between the

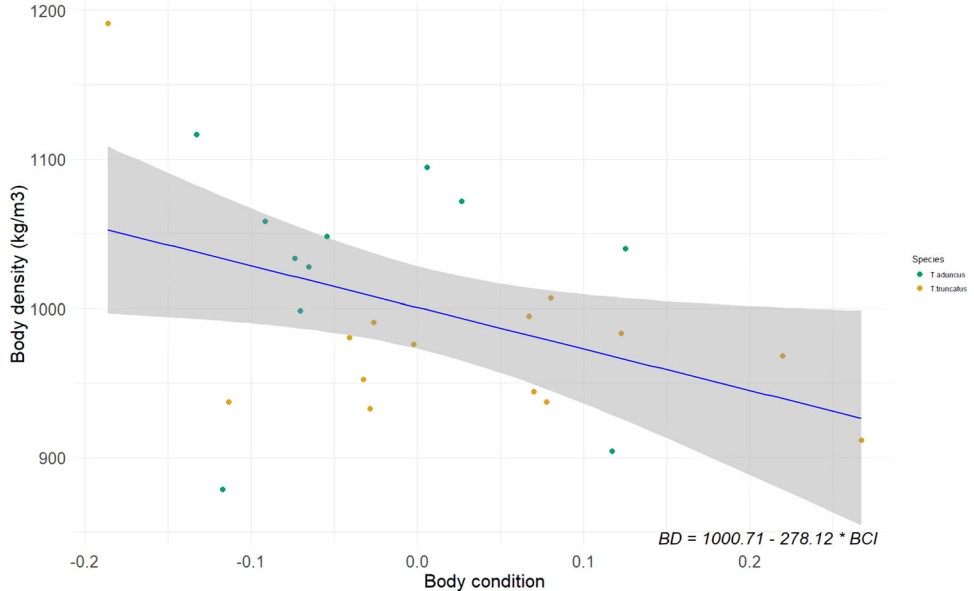

*BD = 1000.71 - 278.12 * BCI*

**Figure 4. Linear relationship between body density (BD) and body condition index (BCI) of bottlenose dolphins**
The blue line represents the fitted values of the linear model with the shaded area showing the 95% confidence interval. *N* = 25.

measured BM and BL on the log-log scale ($F_{(1, 23)} = 164$, $P < 0.001$, $R^2 = 0.87$; Fig. 3*C* and *D*), showing that BM also increases exponentially with length:

$$\log\left(BM_{exp,i}\right) = 3.01 + 2.44 * \log\left(BL_i\right) \qquad (10)$$

The comparison between $BCI_{BV}$ and $BCI_{BM}$ revealed a strong positive correlation ($F_{(1, 23)} = 42.8$, $P < 0.001$, $R^2 = 0.64$, intercept = $0.0014 \pm 0.013$, slope = $0.91 \pm 0.014$; Fig. 3*E*), showing that $BCI_{BV}$ closely resembles the mass-based body condition index ($BCI_{mass}$).

**Assessing the accuracy of the body mass model.** The estimated BM of the stationary dolphins, calculated using BL, W and H (predicted from W), showed a strong correlation to the true BM ($F_{(1, 23)} = 461$, $P < 0.001$, $R^2 = 0.95$, intercept = $-0.49 \pm 9.1$, slope = $1.01 \pm 0.047$; Fig. 6*A*). The mean measurement error was 6.42% (SD = 11.95, range = $-30.39$–21.14 kg). Similarly the estimated BM of freely swimming dolphins, based on their BL, Ws and Hs (predicted from Ws), was strongly correlated with the true BM ($F_{(1, 23)} = 543$, $P < 0.001$, $R^2 = 0.96$, intercept = $-17.58 \pm 9.0$, slope = $1.09 \pm 0.047$; Fig. 6*B*). Here mean measurement error was 6.4% (SD = 11.93, range = $-19.52$ to 24.24 kg). These results indicate that BM estimates measured from stationary and freely swimming dolphins closely align with their true BM.

The equation used to estimate BM is based on the product of BV and BD. In this case, BD was replaced using Eqn 6b, which depends on body condition. BCI, in turn, was substituted using Eqn 9. This resulted in the following equation:

$$BM_{Pred} = BV_{obs}$$

$$* \left(1000.71 - 278.12 * \frac{BV_{obs} - e^{-4.026+2.5959 \times \log(BL)}}{e^{-4.026+2.5959 \times \log(BL)}},\right)$$

$$(7c)$$

## Discussion

This study provides a comprehensive validation of UAV-based photogrammetry for measuring cetacean body morphometrics, volume, condition and mass, using bottlenose dolphins (*Tursiops* spp.), a small cetacean, as a model taxon. Our results showed high consistency among methods for estimating BL, W and girth, with only minor differences between hand measurements of dolphins, and UAV measurements of stationary, and freely swimming dolphins. Notably UAV measurements exhibited similar accuracy yet higher precision than hand measurements. Additionally, we validated the volume-based BCI developed by Christiansen et al. (2018, 2019) as a measure of an animal's energy reserves by showing a strong positive correlation with the mass-derived BCI, demonstrating that variation in BV provides an accurate reflection of body condition in dolphins. Furthermore, we showed that UAV-derived BM estimates for both stationary and freely swimming dolphins closely matched true BM. This validation underscores the potential of UAV photogrammetry as a non-invasive, accurate and

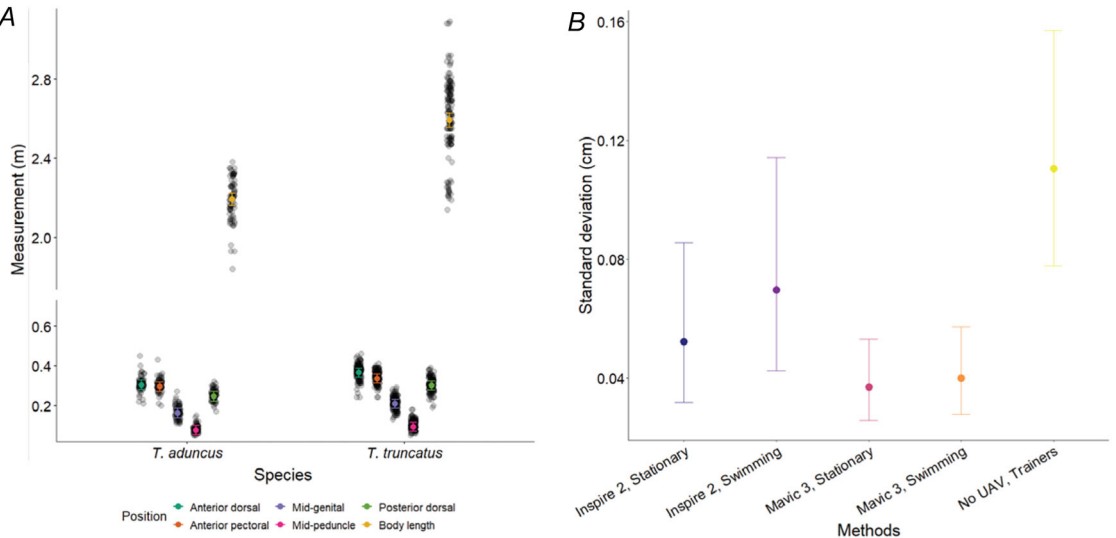

**Figure 5. Comparison of body length and body width measurements**
*A*, body length (BL) and body width (W) at five body sites of *Turiops truncatus* and *Turiops aduncus*. Black points represent the individual measurements, whereas coloured points and error bars show the model-predicted means and associated 95% confidence interval for each species and body site. *B*, SD in estimated BL and W of bottlenose dolphins from measurements collected manually by trainers and from unoccupied aerial vehicle (UAV)-derived measurements using two UAV types (Inspire 2 and Mavic 3) under two conditions (Stationary and Swimming), respectively. *N* = 1345.

precise tool for monitoring body size, condition and mass in wild cetaceans. Given the growing use of UAV-based photogrammetry in studying cetaceans' physiology and bioenergetics, this study plays a crucial role in affirming the method's validity.

## Validation of UAV-derived morphometric measurements

Our validation analyses confirmed the robustness of UAV-based photogrammetry across different measurement conditions. Inter-observer reliability was consistently high for both manual and UAV-based measurements, whether animals were stationary or freely swimming. Moreover, the strong agreement between methods demonstrated that UAV-based photogrammetry yields results equivalent to traditional manual measurements. The location-dispersion GLMM further supported these findings, revealing negligible differences between UAV- and hand-derived measurements while showing that UAV-based approaches were generally more consistent. In particular, the UAV-based measurements exhibited significantly lower variability than manual measurements, highlighting the enhanced precision of UAV photogrammetry. This improvement likely stems from the bird's eye view of UAV images, which offer a fixed reference that minimizes errors and ensures more consistent estimates across repeated assessments. By contrast, manuals measurements with a tape measure are more susceptible to variability, possibly due to the trainers' ground-level perspective and the difficulty in repeatedly locating the same body site while the animal is moving.

Previous studies have also demonstrated the accuracy of UAV photogrammetry. Dawson et al. (2017) reported very small errors when measuring a floating target, whereas studies on wild mysticetes found similarly low variation when using scaling objects (Burnett et al., 2019; Christiansen et al., 2018; Durban et al., 2016). Although these efforts advanced the field, our study adds a novel contribution by validating UAV-based morphometric measurements of living animals. As aerial photogrammetry is ultimately applied to free-ranging individuals, demonstrating that UAV-derived measurements reliably reflect the actual size and shape of living animals represents an important step forward in this field of research.

Having established the accuracy of UAV-derived measurements, we then explored their application to describing body shape. This study provides a comprehensive description of the cross-sectional body shape (height:width HW ratio) of bottlenose dolphins. We found consistent vertical elongation (HW > 1) along the body, with more pronounced lateral flattening near

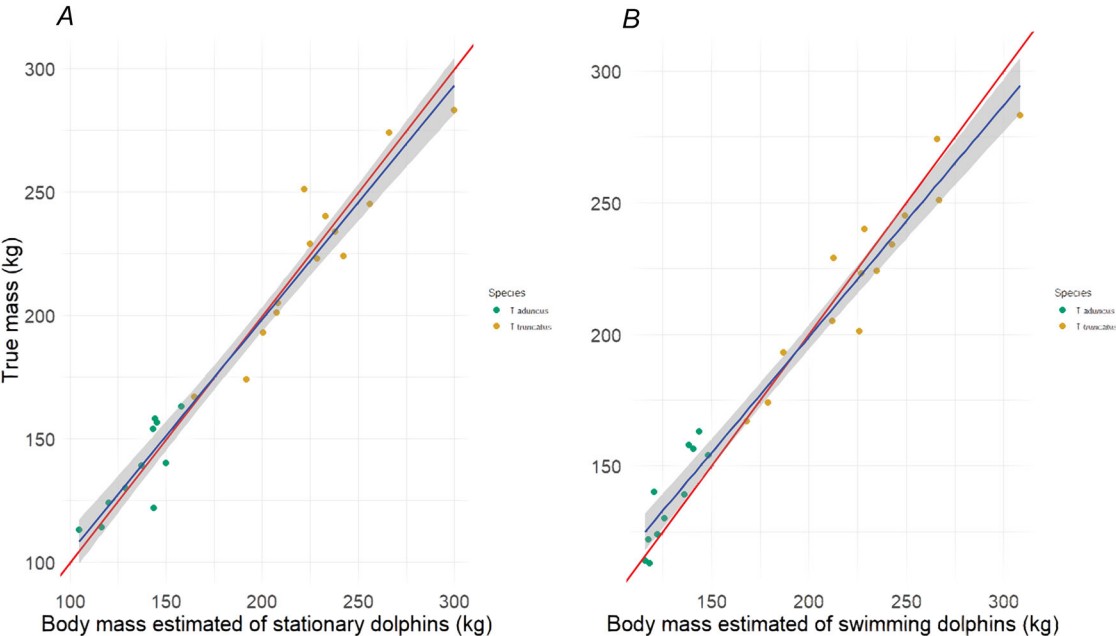

**Figure 6. Linear relationships between true (manually measured) and unoccupied aerial vehicle (UAV)-estimated body mass**
*A*, linear relationship between manually measured body mass (BM) and UAV-estimated BM when animals were stationary (intercept = −0.49 ± 9.1, slope = 1.01 ± 0.047). *B*, linear relationship between manually measured BM and UAV-estimated BM when animals were swimming (intercept = −17.58 ± 9.0, slope = 1.09 ± 0.047). The red lines represent a 1:1 relationship, whereas the blue lines represent the fitted values of the linear model with the shaded area showing the 95% confidence interval. *N* = 25.

the tail and greater individual variability in the middle section, likely reflecting the most metabolically active region. Similar elongation has been reported in sperm whales (*Physeter macrocephalus;* Glarou et al., 2023), short-finned pilot whales (*Globicephala macrorhynchus;* Arranz et al., 2022) and East Asian finless porpoises (*Neophocaena asiaeorientalis sunameri*; Zhang et al., 2023). Lateral flattening is thought to enhance laminar flow and reduce drag (Nesteruk, 2019), providing an advantage when pursuing fast prey such as certain fish and cephalopods (Aguilar de Soto et al., 2008; Clarke & Clarke, 1997). Comparable variability in the middle body region has been observed in short-finned pilot whales (Noren et al., 2019), beluga (Cornick et al., 2016), harbour porpoises (*Phocoena phocoena*; Koopman 1998) and common dolphins (Joblon et al., 2014) and aligns with the evidence that metabolically active regions occur between 40% and 80% of BL (Christiansen et al., 2016, 2018, 2024). In contrast, many mysticetes exhibit a more circular anterior cross-section that flattens posteriorly (Christiansen et al., 2019; Christiansen, Sprogis et al., 2020), a shape associated with slower, bulk-feeding strategies (Pivorunas, 1979; Watkins & Schevill, 1979) where speed and drag reduction are less critical (Simon et al., 2009; Werth, 2004).

### Validation of UAV-derived volume-based body condition index

For terrestrial animals, BCIs are typically derived by standardizing BM against BL, with length serving as a proxy for structural size (Green, 2001; Peig & Green, 2009; Schulte-Hostedde et al., 2005). This approach assumes that variation in length primarily reflects skeletal size, whereas residual variation in mass reflects energy reserves. Although one- and two-dimensional metrics can serve as useful proxies of condition in marine mammals (Bierlich et al., 2021), they do not necessarily scale directly with BM. By contrast, BV correlated strongly with mass, and our results showed that volume-based indices closely matched mass-based estimates, underscoring the accuracy and reliability of this method for assessing body condition in dolphins and other marine mammals. Previous validation work has shown that 3D and elliptical models derived from porpoise carcasses can accurately reproduce true volumetric estimates (Zhang et al., 2023). Similarly, Irschick et al. (2020) reported a close agreement between volumetric models and direct measurements in harbour porpoises, whereas Shero et al. (2014) demonstrated that an elliptical truncated cone model improved BM estimates in live Weddell seals (*Leptonychotes weddellii*). Our study builds on this work by extending validation efforts to a larger sample of live animals and by directly comparing volume- and mass-based body condition metrics. This is important for marine mammal research, where photogrammetry offers a logistically feasible, cost-effective and non-invasive method for assessing body condition. By ground-truthing this approach, we strengthen its application in ecological and bioenergetic studies and provide a crucial link to the broader body of research on body condition metrics traditionally based on mass in terrestrial systems (Green, 2001; Peig & Green, 2009; Schulte-Hostedde et al., 2005). This validation enables researchers to confidently use volume-based photogrammetry to investigate bioenergetics and physiology in free-ranging marine mammals.

### Validation of UAV-derived body mass estimation

Accurately estimating BM is critical in ecological and physiological research, as this parameter directly influences metabolic rate, energy requirements and hence food intake (Lauder, 1981). Allometric scaling equations that link BM to metabolism have long been used to predict energy demands and to estimate ecosystem carrying capacities when direct measurements are not feasible (Brown et al., 2004; Nagy, 2001). The consistency observed in mass estimates of both stationary and freely swimming subjects underscores the effectiveness of UAV photogrammetry as a reliable, non-invasive method for obtaining morphometric data in natural settings. In our study, UAV-derived estimates of BM closely matched actual BM with an average error of about 6.4%. Comparable levels of accuracy have been reported in other species. In Galapagos sea lions (*Zalophus wollebaeki*), photogrammetry achieved BM estimates within ±6.8% for males and ±14.5% for females compared to direct measurements (Meise et al., 2014), whereas in Weddell seals, three-dimensional photogrammetry produced mass estimates that were statistically indistinguishable from true values, averaging 345–346 kg compared to an actual mean of 346 ± 73 kg (Beltran et al., 2018). Beyond marine mammals, similar accuracy has been documented in terrestrial species (Postma et al., 2015). By validating UAV photogrammetry for estimating BM in odontocetes, our study provides empirical evidence that such estimates closely approximate the true BM of living animals, supported by the strong agreement between photogrammetric estimates and manual mass measurements.

### Study limitations

Measurement errors due to water distortion or variations in body posture can affect photogrammetric measurements (Christiansen et al., 2018). We attempted to mitigate these factors by selecting images with minimal water spray, clear body contours and straight post-

ures, resulting in highly accurate estimates. For the free-swimming condition, we recorded an average 8 min of video per individual and followed each animal for approximately six surfacing events. From these events, the single best image was selected for measurement. For field applications, where conditions are less controlled, we recommend recording at least ten surfacing events per individual, whenever possible, and then using the best single image for morphometric measurements. The close agreement between UAV-derived and manual measurements indicates that distortion was not a major source of error in our dataset.

In addition, because all measurements were collected at the same time of year, we were unable to capture seasonal changes in body condition (range: −0.186 to 0.267). Repeated measurements across different seasons could provide a more complete picture of the morphological range that dolphins naturally exhibit over the course of the year, as documented in other populations, such as Tamanend's bottlenose dolphins (*T. erebennus*; Perkins-Taylor et al. 2024) and common bottlenose dolphins (Adamczak et al., 2021).

Our height:width (HW) ratios were derived from a dataset comprising mostly of adults and a few larger juveniles ($n = 4$; body length range: 1.96–2.99 m). The limited access to individuals from different age classes may not fully represent the ontogenetic variation in body shape, as allometric relationships describing how body proportions scale with length often shift during growth (Christiansen et al., 2019; Christiansen, Sprogis et al., 2020). For instance, differences in HW ratios have been reported between calves and adults in right and humpback whales (Christiansen et al., 2019; Christiansen, Sprogis et al., 2020). Similar patterns may exist in bottlenose dolphins. Mallette et al. (2016) showed clear ontogenetic changes in body shape and composition across age classes in bottlenose dolphins, whereas Kurihara and Oda (2009) demonstrated allometric changes in skull morphology. Incorporating younger and smaller individuals in future photogrammetric studies will therefore be essential for capturing the full range of morphometric development in bottlenose dolphins.

Aerial photogrammetry studies showed that late-pregnant females exhibit substantial expansion in width across the mid-body region in both toothed (30%–60% of BL from the rostrum; Cheney et al., 2022) and baleen whales (50%–65% of BL from the rostrum; Christiansen et al., 2022; van Aswegen et al., 2025), reflecting the physical space occupied by the foetus, placenta and associated fluids. Whether this increase in width is matched by a proportional increase in dorso-ventral height remains unknown, as lateral images of late-pregnant individuals are currently unavailable.

For non-pregnant adults, we found no difference in HW ratios between males and females (Table S2), suggesting similar cross-sectional body shape. This has important practical implications, as dolphins cannot be sexed reliably from aerial photographs, making sex-specific HW ratios difficult to apply in the field.

## Ecological implications

The validation of this method carries broad implications across ecological and evolutionary research. UAV photogrammetry is increasingly being used to obtain morphometric information from free-ranging cetaceans, enabling the assessment of age structure, demographic patterns and population trajectories in the wild. For example, it has been applied to monitor a critically endangered bottlenose dolphin population in the Gulf of Ambracia, Greece, where precise BL measurements informed age distribution and provided insights into population stability (Vivier et al., 2024). Healthy populations typically show a balanced representation of calves, juveniles and adults (Gamelon et al., 2016), whereas deviations from this distribution can signal population growth or decline (Coulson et al., 2005; Jackson et al., 2020; Jones et al., 2018). Monitoring shifts in body size over time can therefore provide early warning signals of population-level stressors.

Beyond structural traits, body condition provides important information on the energetic status of individuals and their ability to meet the demands of growth, reproduction and survival (Anderson-Teixeira et al., 2009; Kleiber, 1947). In particular, maternal condition has strong implications for reproduction, as females in good condition tend to produce larger, more viable offspring (Christiansen et al., 2018; Kovacs & Lavigne, 1986; Tauson, 1993). Similar relationships between maternal condition and offspring survival have been documented in mysticetes (Christiansen et al., 2014, 2018), pinnipeds (Kovacs & Lavigne, 1986; McDonald et al., 2008), ungulates (Parker et al., 2009) and birds (Lorentsen, 1996).

In males, good body condition enhances reproductive success by sustaining energetically costly mating behaviours, as shown in species ranging from bullfrogs to pinnipeds and primates (Higham et al., 2011; Judge & Brooks, 2001; Lidgard et al., 2005). In Shark Bay, male bottlenose dolphins form alliances that engage in prolonged conflicts over females, sometimes guarding them for weeks (Connor et al., 2022). Such strategies require substantial energy, suggesting that males in better condition may achieve greater mating success.

## Management and implications

At the population level, body condition also reflects nutritional health and demographic trends, making it

a critical tool for wildlife monitoring and conservation (Christiansen, Dawson et al., 2020; Rolland et al., 2016; Stevenson & Woods, 2006). Marine ecosystems are rapidly changing under the influence of climate and other anthropogenic pressures (Crain et al., 2008; Doney et al., 2012; Hawkins et al., 2017; Hoegh-Guldberg & Bruno, 2010), which can disrupt prey availability, alter community structures and reduce reproductive success (Bhagarathi et al., 2024; Cubero-Pardo et al., 2011). These ecosystem-level shifts can often be detected as changes in animal's body condition. For example, marine heatwave events have been associated with marked declines in body condition in blue (Wachtendonk et al., 2022) and grey whales (Stewart et al., 2023), and similar stressors have reduced both survival and calving rates in Shark Bay's Indo-Pacific bottlenose dolphins (Wild et al., 2019). The validation of non-invasive methods for remotely estimating BM and condition therefore strengthens confidence in using these approaches to assess how environmental disruptions influence population health.

The validation of UAV photogrammetry for accurately measuring body size, condition and mass in bottlenose dolphins represents an important advancement for conservation and management, especially for species where traditional health assessment is impractical or impossible. For instance, there is currently no safe, routine method to capture and restrain baleen whales for physiological sampling, and there are no facilities capable of housing species longer than 8 m (i.e. such as killer whales, *Orcinus orca*). By testing the accuracy of a framework for estimating key morphological and energetic parameters without handling animals, our study offers a foundation for extending UAV-based photogrammetry to other cetaceans where direct validation is not feasible (Hunt et al., 2013). Given the role of marine mammals as sentinel species (Bossart, 2011; Moore, 2008), integrating these remote measurements into long-term monitoring programmes will allow managers to detect emerging stressors earlier, evaluate population responses to environmental change and support species management and conservation in a rapidly shifting ocean.

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

## Additional information

### Data availability statement

All data supporting the findings of this study are publicly available in the Zenodo Repository: https://doi.org/10.5281/zenodo.17475254.

### Competing interests

The authors declare no competing interests that are relevant to the content of this article.

### Author contributions

R.C. conceived and designed the study with the input from S.J.A., F.C. and M.K., whereas M.K. and W.P. obtained funding for the data collection. R.C., B.M. and M.R.B collected the data with the logistic support of S.J.A. and W.P., and R.C. processed the data. R.C., F.C. and E.P.W. analysed the data, and R.C. wrote the manuscript. All authors contributed to subsequent drafts and gave final approval for publication.

### Funding

This study was supported by grants from the A.H. Schultz Foundation (Dept. of Evolutionary Anthropology, University

of Zurich) and the Swiss National Science Foundation (310030_204974 to M.K.).

## Acknowledgements

We thank Sea World Australia for hosting us and granting access to its animals for data collection. We are also grateful to the dolphin trainers for their dedicated work in training the animals and collecting hand measurements, which made this research possible. We would like to further thank all members of the Evolutionary Genetics Group and the Department of Evolutionary Anthropology at the University of Zurich for their support. Finally we would like to thank the reviewers for their feedback on this manuscript.

Open access publishing facilitated by Universitat Zurich, as part of the Wiley - Universitat Zurich agreement via the Consortium Of Swiss Academic Libraries.

## Keywords

body condition, body mass, bottlenose dolphins, drone photogrammetry, non-invasive morphometrics, validation

## Supporting information

Additional supporting information can be found online in the Supporting Information section at the end of the HTML view of the article. Supporting information files available:

**Peer Review History**
**Fig. S1**
**Fig. S2**
**Table S1**
**Table S2**
**Table S3**
**Table S4**

