## [Peer Review History · The Journal of Physiology]

Validation of aerial photogrammetry methods to measure body size, condition, and mass in small cetaceans

Riccardo Ciciarella, Erik P. Willems, Benjamin Markham, Manuela R. Bizzozzero, Wayne Phillips, Simon J. Allen, Michael Krützen, and Fredrik Christiansen

DOI: 10.1113/JP290419

Corresponding author(s): Riccardo Ciciarella (riccardo.ciciarella@iea.uzh.ch)

The following individual(s) involved in review of this submission have agreed to reveal their identity: Lucas L. de Oliveira (Referee #2)

Review Timeline:

Submission Date:	30-Oct-2025
Editorial Decision:	24-Nov-2025
Revision Received:	22-Dec-2025
Accepted:	07-Jan-2026

Senior Editor: Eleonora Grandi

Reviewing Editor: Janna Morrison

Transaction Report:

Dear Dr Ciciarella,

Re: JP-RP-2025-290419 "Validation of aerial photogrammetry methods to measure body size, condition, and mass in small cetaceans" by Riccardo Ciciarella, Erik P. Willems, Benjamin Markham, Manuela R. Bizzozzero, Wayne Phillips, Simon J. Allen, Michael Krützen, and Fredrik Christiansen

Thank you for submitting your manuscript to The Journal of Physiology. It has been assessed by a Reviewing Editor and by 2 expert referees and we are pleased to tell you that it is acceptable for publication following satisfactory revision.

Please address all the points raised and incorporate all requested revisions or explain in your Response to Referees why a change has not been made. We hope you will find the comments helpful and that you will be able to return your revised manuscript within 4 weeks.

The Editor also recommends that your manuscript is revised as a Techniques article, as opposed to Research, and therefore we will be changing it to this new format within the system. For full information on this article type, please see this webpage: https://jp.msubmit.net/cgi-bin/main.plex?form_type=display_requirements#techniques. If you have any questions about this change, or require longer to make the revisions, please contact journal staff: jp@physoc.org.

REVISION CHECKLIST:

We look forward to receiving your revised submission.

Yours sincerely,

Eleonora Grandi
Senior Editor
The Journal of Physiology

REQUIRED ITEMS

- Please upload separate high-quality figure files via the submission form.

EDITOR COMMENTS

Reviewing Editor:

Comments to the Author:

Thank you for this interesting manuscript. Great to see strong validation of a method for monitoring dolphins in the wild without interaction.

Please provide information about food and housing for the dolphins at Seaworld. Did the UWA ethics cover the work at Seaworld?

Senior Editor:

Comments to the Author:

I concur with the Reviewing Editor's recommendation.

REFEREE COMMENTS

Referee #1:

This manuscript is very well written, clear and concise. The study design and methodology are sound, and the analyses are appropriate for the stated aims and objectives. I find the paper to be both interesting and valuable, particularly for marine mammal researchers, and I believe it would represent a meaningful contribution to the use of UAVs in conservation efforts.

My main suggestion is that, while the value of the study is clear, the manuscript could benefit from a more focused and specific discussion of how it contributes to existing and future research. Rather than repeating this point throughout the manuscript, I recommend incorporating specific examples in a more targeted manner, perhaps in the "Management and Implications" section toward the end. Additionally, the manuscript alternates between the terms "drone" and "UAV." For consistency, I suggest choosing one term and using it consistently throughout (I personally prefer "UAV").

The paper is very well done so I only have a few questions, suggestions and points of clarification:

Line 65: Suggest adding in "...predictor of age class...." Age groups are mentioned later but important to acknowledge can't estimate exact age.

Line 67-68: Suggest removing reference to fish. As quite specific to this group e.g. egg size and the rest of the intro, I think, justifies why body size is important to study.

Line 76-84: Clearly important but suggest one example of this is sufficient as this study didn't look at pregnancy or reproduction.

Line 104: First mention of captured animals but clearly one way to get these types of measurements. Suggest adding to start of paragraph - e.g. "...collected during whaling operations or from stranded or captured individuals" For example these type of measurements are regularly collected during health assessments in Sarasota Florida.

Line 104: Agree for strandings/whaling but for can have long term monitoring during health assessment programmes.

Line 106-107: Totally agree for strandings but not necessarily for captures or whaling.

Line 109-110: Totally agree!

Line 111-117: Suggest integrating this paragraph into the last paragraph especially as you mention capture there. E.g. There is logistical challenges but can get data from strandings, capture, whaling. But cons of these methods. So UAV important.

Line 112-114: Also even for small cetaceans in some areas/species capture or restraining is isn't possible as is impractical or unsafe e.g. deep water, larger small cetacean, cost, ethics, etc.

Line 183: ".....five straight-line measurements" difficult to understand. The figure helps with this but suggest adding "....five straight line measurements of body width and height at five....."

Line 188: Assume each trainer measured each individual dolphin once? Would be good to clarify.

Line 189: "....extracted measures....." is a little unclear.

Line 192-193: Three measurers who measured each individual once? Would be good to ensure clear.

Line 388: Equation 6b is based on equation 6a - be good to clarify as I initially wondered where the 100.71 and 278.12 came from.

Line 401-402. Suggest specifying this is between species as think unclear until next sentence.

Line 407: Expand on what was the significant difference in measurements between mavic 3 drone and swimming condition.

Line 411: May not be an issue but be good to expand briefly on this significant difference e.g. between mavic 3 and hand measurements or variation within mavic 3?

Line 424: BCIBM was previously BCImass? Good to keep all these terms consistent for clarity.

Line 519: remove "particularly" as not needed. It is important.

Line 524-526: Need specific example where population dynamics comes in

Line 532: remove "remarkable". Subjective term.

Line 543-546: This is quite a strong statement. Suggest providing examples to help justify it.

Line 548-549: Suggest removing this sentence and going straight into the limitations. Since the title already indicates the focus, this is not necessary

Line 550-552: For a field setting it would be useful to know how many hours of video were taken and how many images were useful for measuring.

Line 557: Interesting to think about pregnancy and how that would change height:width ratio and body condition/mass. Something to add? Also was there any difference between males and females that could/should be investigated?

Line 560: Were you able to see any different between the juveniles and adults? Even if not significant with small sample size.

Line 574-577: Repeated from introduction

Line 572-587: Suggest condensing this paragraph, as the focus of the study is primarily on volume, mass, and condition, rather than body length.

Line 588-601: As above, important points, but suggest could be reduced as although the results from this study pave the way for future studies on maternal condition, this study didn't investigate that.

Line 607-610: Fits in more clearly to the next paragraph

Referee #2:

I would like to congratulate the authors on the production of the manuscript entitled "Validation of aerial photogrammetry methods to measure body size, condition, and mass in small cetaceans".

This is a valuable manuscript that addresses a critical need in marine mammal science. The authors have conducted a rigorous, well-designed study to validate UAV-based photogrammetry for assessing the morphometrics of small cetaceans. The work is thorough, the data is robust, and the conclusions are exceptionally well-supported.

The impact of this research is significant. It provides robust, empirical validation for a non-invasive, and accurate tool that is vital for the future of cetacean ecology and conservation, demonstrating that UAV-derived data can be reliable, allowing the widespread adoption of this method in monitoring body condition, nutritional stress, and reproductive success in free-living marine mammal populations where direct handling is impractical or impossible.

One of the major findings of the manuscript was that intra-method comparison analysis showed that replicas of measurements obtained by drones are more precise than measurements obtained by different handlers.

END OF COMMENTS

Riccardo Cicciarella, MSc
University of Zurich
Department of Anthropology
Evolutionary Genetics Group
Zurich, 8057, Switzerland
E-mail: riccardo.cicciarella@iea.uzh.ch

Zurich, 22nd December 2025

The Journal of Physiology

Re: Submission of the revised manuscript for evaluation: “Validation of aerial photogrammetry methods to measure body size, condition, and mass in small cetaceans”

(ID: JP-RP-2025-290419)

Authors: Riccardo Cicciarella¹, Erik P. Willems¹, Benjamin Markham², Manuela R. Bizzozzero¹, Wayne Phillips², Simon J. Allen^{1,3,4}, Michael Krützen^{1†}, Fredrik Christiansen^{5†}

¹ Evolutionary Genetics Group, Department of Evolutionary Anthropology, University of Zurich, 8057 Zurich, Switzerland.

² Marine Sciences, Sea World Foundation Australia, Main Beach QLD 4217, Australia.

³ School of Biological Sciences, University of Bristol, Bristol, BS8 1TQ, UK.

⁴ School of Biological Sciences, Oceans Institute, University of Western Australia, Crawley, WA 6009, Australia.

⁵ Marine Mammal Research, Department of Ecoscience, Aarhus University, 4000, Roskilde, Denmark.

* Corresponding author: riccardo.cicciarella@iea.uzh.ch

† These authors contributed equally to this work

Dear Dr. Eleonora Grandi,

We would like to sincerely thank you, the Reviewing Editor, and the two referees for your time and constructive feedback provided on our manuscript. We have carefully revised the manuscript in response to these comments and are resubmitting to *The Journal of Physiology*. Detailed responses to each point are indicated below.

In revising the manuscript, we addressed all comments from the Reviewing Editor and referees. The manuscript has been substantially improved by clarifying methodological details, strengthening the structure of the Introduction and Discussion, and streamlining sections where repetition had been noted. We incorporated additional methodological information, refined the presentation of statistical analyses, and ensured consistency in terminology throughout. In line with the referees' recommendations, we expanded the "Management and implications" section to include more targeted examples, illustrating how the validated photogrammetry methods can inform ecological monitoring and conservation. We have also condensed sections where the focus drifted from the central aims of the study, removed redundancies, and integrated new analyses where requested (e.g. assessing sex effects on HW ratios).

Please find below the referees' comments and our responses.

In the following, each referee comment is highlighted in grey boxes with red text, followed by our response. In the manuscript, changes are highlighted in red font.

Thank you again, and we hope the revised manuscript meets with your approval.

Best regards,

Riccardo on behalf of all co-authors.

Reviewing Editor:

(1) Reviewing Editor – General comment – *Please provide information about food and housing for the dolphins at Seaworld. Did the UWA ethics cover the work at Seaworld?*

Response to general comment: Yes, the manuscript will benefit from including this information, and we have added it to the methods section. We also revised the “Ethical approval” section to ensure that it reflects the permits and institutional oversight that applied to the activities conducted at Sea World.

*Lines 161-168: “Morphometric data were collected on 14 *Tursiops truncatus* (n = 8 females and n = 6 males) and eleven *T. aduncus* (n = 5 females and n = 6 males) under human care at the Sea World in Gold Coast, Queensland, Australia. The dolphins were housed in lagoons with sandy (natural) substrates and supplied with water sourced from natural waterways. The dolphins were maintained in both same- and mixed-sex groups. Dolphins were fed a seasonally adjusted, individually tailored diet consisting of a minimum of five different fish species daily, all sourced through human-grade fish brokers. Rations varied according to individual energetic requirements and ranged approximately from 5.5 to 15 kg of fish per animal per day. Commonly fed fish species are submitted for nutritional analysis twice annually to ensure accurate characterization of the diet’s nutritional composition.”*

Lines 149-156: “All work involving dolphins at Sea World took place within the facility’s established animal care, husbandry, and veterinary programs. Data were collected in coordination with Sea World and in accordance with their operational guidelines. Morphometric measurements were provided by Sea World staff as part of their ongoing health practices and were obtained under the facility’s Exhibition Animals Permit (PRID000821), issued by the Queensland Department of Environment and Science. No procedures required handling beyond standard husbandry practices.”

Referee 1:

(1) Referee 1 – General comment – *My main suggestion is that, while the value of the study is clear, the manuscript could benefit from a more focused and specific discussion of how it contributes to existing and future research. Rather than repeating this point throughout the manuscript, I recommend incorporating specific examples in a more targeted manner, perhaps in the “Management and Implications” section toward the end. Additionally, the manuscript alternates between terms “drone” and “UAV.” For consistency, I suggest choosing one term and using it consistently throughout (I personally prefer “UAV”).*

Response to general comment 1: We appreciate the referee’s feedback. We changed the term “drone” to “UAV” throughout the manuscript. We also streamlined the Discussion to reduce repetition and ensure that statements about the broader relevance of our findings are more focused. We revised the “Management and implications” section to incorporate specific examples illustrating how the validated methods can be applied in ecological monitoring and conservation management. These examples now outline how reliable UAV-based estimated of body condition can support assessment of nutritional stress, populations responses to environmental change, and the monitoring of species for which direct health evaluations are not feasible. The “Management and implication” section reads as follow:

Lines 636-662: “At the population level, body condition also reflects nutritional health and demographic trends, making it a critical tool for wildlife monitoring and conservation (Stevenson & Woods, 2006; Rolland et al., 2016; Christiansen et al., 2020a). Marine ecosystems are rapidly changing under the influence of climate and other anthropogenic pressures (Crain et al., 2008; Hoegh-Guldberg & Bruno, 2010; Doney et al., 2012; Hawkins et al., 2017), which can disrupt prey availability, alter community structures, and reduce reproductive success (Cubero-Pardo et al., 2011; Bhagarathi et al., 2024). These ecosystem-level shifts can often be detected as changes in animal’s body condition. For example, marine heatwave events have been associated with marked declines in body condition in blue (Wachtendonk et al., 2022) and gray whales (Stewart et al., 2023), and similar stressors have reduced both survival and calving rates in Shark Bay’s Indo-Pacific bottlenose dolphins (Wild et al., 2019). The validation of non-invasive methods for remotely estimating body mass and condition therefore strengthens confidence in using these approaches to assess how environmental disruptions influence population health.

*The validation of UAV photogrammetry for accurately measuring body size, condition, and mass in bottlenose dolphins represents an important advancement for conservation and management, especially for species where traditional health assessment are impractical or impossible. For instance, there is currently no safe, routine method to capture and restrain baleen whales for physiological sampling, and no facilities capable of housing species longer than 8 m (i.e. such as killer whales, *Orcinus orca*). By testing the accuracy of a framework for estimating key morphological and energetic parameters without handling animals, our study offers a foundation for extending UAV-based photogrammetry to other cetaceans where direct validation is not feasible (Hunt et al., 2013). Given the role of marine mammals as sentinel species (Moore, 2008; Bossart, 2011), integrating these remote measurements into*

long-term monitoring programmes will allow managers to detect emerging stressors earlier, evaluate population responses to environmental change, and support species management and conservation in a rapidly shifting ocean.”

(2) Referee 1 – Comment (Line 65) – Suggest adding in “...predictor of age class...” Age groups are mentioned later but important to acknowledge can’t estimate exact age.

Response to comment 2: We thank the referee for noting this. We edited the text in a way that acknowledges the fact we cannot estimate exact age, rather only the age class.

Line 64-67: “Body length, in particular, serves as a reliable predictor of age class (Vivier et al., 2023), enabling researchers to classify individuals into age groups and assess key demographic parameters such as survival and reproductive rates (Holmes & York, 2003).”

(3) Referee 1 – Comment (Line 67-68) – Suggest removing reference to fish. As quite specific to this group e.g. egg size and the rest of the intro, I think, justifies why body size is important.

Response to comment 3: We are grateful to the referee for pointing this out. We removed the examples and reference about fish.

Line 67-68: “Body length can also serve as a proxy for growth rate in many vertebrates (Davis et al., 2008; Mott et al., 2010; Andrialovanirina et al., 2020).”

(4) Referee 1 – Comment (Line 76-84) – Clearly important but suggest one example of this is sufficient as this study didn’t look at pregnancy or reproduction.

Response to comment 4: We value the referee’s recommendation and deleted two of the three examples. The paragraph now reads as follows:

*Line 71-79: “Body condition provides insights into the energetic demands of life functions including body maintenance, growth, mating and reproduction (Kleiber, 1947; Anderson-Teixeira et al., 2009). In females, body condition is often positively related to reproductive success, including birth size, offspring condition, growth rate, and survival (Tauson, 1993; Taillon et al., 2012; Christiansen et al., 2018). Body condition can also serve as an indicator of the energetic costs of gestation and lactation. In Southern right whales (*Eubalaena australis*), for example, lactating females lose about 25% of their body volume over a three-month breeding season, with body condition loss being positively related to calf growth rate (Christiansen et al., 2018).”*

(5) Referee 1 – Comment (Line 104) – *First mention of captured animals but clearly one way to get these types of measurements. Suggest adding to start of paragraph – e.g. “...collected during whaling operations or from stranded or captured individuals” For example these type of measurements are regularly collected during health assessments in Sarasota Florida.*

Response to comment 5: We appreciate the referee’s helpful comment. We modified the sentence so to mention captured animals from the beginning as suggested by the referee.

Line 90-92: “In cetaceans, body size and condition are commonly estimated by taking morphometric measurements, collected during whaling operations or from stranded or temporarily restrained individuals (Lockyer, 1981; Fortune et al. 2012; Barratclough et al., 2019).”

(6) Referee 1 – Comment (Line 104) – *Agree for strandings/whaling but for can have long term monitoring during health assessment programmes.*

Response to comment 6: We thank the referee for highlighting this aspect. We included a reference to long term monitoring for health assessment programs.

Line 98-101: “While these methods have provided valuable insights and are used in health assessment programmes for population-level monitoring (Wells et al., 2004), they rely on captured or stranded animals and therefore preclude long-term monitoring of small, remote, or endangered populations.”

(7) Referee 1 – Comment (Line 106-107) – *Totally agree for strandings but not necessarily for captures of whaling.*

Response to comment 7: We are grateful for the referee’s suggestion. We specified that post-mortem distortion applies only to strandings.

Line 101-104: “They may also introduce sampling biases towards specific age, reproductive classes, or body condition (Aguilar et al., 1999; Krahn et al., 2004), and post-mortem distortion in stranded animals compromises carcass-based measurements (Moore et al., 2004; Barratclough et al., 2014).

(8) Referee 1 – Comment (Line 109-110) – *Totally agree!*

Response to comment 8: We are happy that the referee finds value in the motivation of our study.

(9) Referee 1 – Comment (Line 111-117) – Suggest integrating this paragraph into the last paragraph especially as you mention capture there. E.g. There is logistical challenges but can get data from strandings, capture, whaling. But cons of these methods. So UAV important.

Response to comment 9: We value the referee’s observation. The paragraph at lines 111-117 was originally integrated into the preceding paragraph. However, after addressing the referee’s subsequent comment, we added several sentences to the 111-117 section, and combining the two paragraphs resulted in an overly long and dense block of text. To maintain clarity and readability, we opted to keep them as separate paragraphs. The revised paragraph is presented as follows:

Line 90-116: “In cetaceans, body size and condition are commonly estimated by taking morphometric measurements, collected during whaling operations or from stranded or captured individuals (Lockyer, 1981; Fortune et al., 2012). These include body length (Lockyer & Waters 1986, Gómez-Campos et al., 2011), which serves as a proxy for structural size. Body condition can be assessed using methods that directly estimate energy reserves, including blubber thickness (Cartee et al., 1995), blubber composition (Aguilar & Borrell, 1990) or relative adipocyte volume (Castrillon et al., 2017), as well as biochemical techniques such as carcass analysis (Iverson et al., 2010) and isotope dilution (Bowen & Iverson, 1998). While these methods have provided valuable insights and are used in health assessment programmes for population-level monitoring (Wells et al., 2004), they rely on captured or stranded animals and therefore preclude long-term monitoring of small, remote, or endangered populations. They may also introduce sampling biases towards specific age, reproductive classes, or body condition (Aguilar et al., 1999; Krahn et al., 2004), and post-mortem distortion in stranded animals compromise carcass-based measurements (Moore et al., 2004; Barratclough et al., 2014). These limitations underscore the need for accurate, non-invasive, and non-lethal methods to assess morphology and condition in wild marine mammals.

Measuring body size and condition in wild cetaceans presents major logistical challenges due to their aquatic lifestyle (Iverson et al., 2010; Ball et al., 2017). Although smaller cetaceans can be captured or restrained, these procedures are logistically complex and induce significant stress (Rojas-Bracho et al., 2019). In many regions, however, capture or restraining is not feasible even for small species due to factors such as water depth or turbidity, unsafe handling conditions, high financial and personnel requirements, or ethical constraints associated with increased risk to both animals and researchers (Norman et al., 2004). The challenges are amplified for species that are elusive, inhabit remote areas, or endangered. For larger marine mammals, the risks associated with close contact often preclude even temporary restraint (Hunt et al., 2013).”

(10) Referee 1 – Comment (Line 112-114) – *Also even for small cetaceans in some areas/species capture or restraining is isn't possible as is impractical or unsafe e.g. deep water, larger small cetaceans, cost, ethics, etc.*

Response to comment 10: We appreciate the referee prompting us to expand on this aspect. We added a sentence that underlines the challenges of capturing small cetaceans.

Line 110-115: “Although smaller cetaceans can be captured or restrained alive, these procedures are logistically complex and induce significant stress (Rojas-Bracho et al., 2019). In many regions, however, capture or restraining is not feasible even for small species due to factors such as water depth or turbidity, unsafe handling conditions, high financial and personnel requirements, or ethical constraints associated with increased risk to both animals and researchers (Norman et al., 2004). The challenges are amplified for species that are elusive, inhabit remote areas, or endangered.”

(11) Referee 1 – Comment (Line 183) – *“.....five straight-line measurements” difficult to understand. The figure helps with this but suggesting adding “...five straight line measurements of body width and height at five.....”*

Response to comment 11: We thank the referee for helping to improve the clarity of this section. We clarified the type of measurements collected:

Line 191-194: “Measurements included body length (BL, distance from tip of rostrum to the end of the tail notch) and five straight-line measurements of body width and height at five defined body positions.”

(12) Referee 1 – Comment (Line 188) – *Assume each trainer measured each individual dolphin once? Would be good to clarify*

Response to comment 12: We are grateful to the referee for suggesting this clarification. We specified that each trainer measured each dolphin once.

Line 196-199: “The trainers took measurements on the dorsal and on one lateral side of each animal. They also measured body girth at the same five body positions. Each trainer measured each individual dolphin once, resulting in three repeated measurement per dolphin.”

(13) Referee 1 – Comment (Line 189) – *“....extracted measures.....” is a little unclear*

Response to comment 13: We value the referee’s feedback. We described in more detailed how the UAV measurements were taken.

Line 200-204: “For the UAV measurements, we selected still frames from the UAV video recordings and used MorphoMetriX to place reference points on the dorsal and one lateral side at the same five body positions. The software calculated distances between reference points in pixels and converted them to real-world measurements using camera metadata and the ground sampling distance (Torres & Bierlich, 2020).”

(14) Referee 1 – Comment (Line 192-193) – Three measurers who measured each individual once? Would be good to ensure clear.

Response to comment 14: We appreciate the referee for noting the need of clarification. We described in more details that three trainers measured each dolphin once.

Line 204-207: “We used images of UAV video recordings when the dolphins were held stationary and when the dolphins were swimming freely near the surface. Three different trainers each measured every individual once, resulting in three repeated measurements per dolphin.”

(15) Referee 1 – Comment (Line 388) – Equation 6b is based on equation 6a – be good to clarify as I initially wondered where 100.71 and 278.12 came from.

Response to comment 15: We thank the referee for highlighting this point. We mentioned that equation 6b is derived from equation 6a.

Line 403-408: “There was a significant negative relationship between BD and BCI ($F_{(1, 23)} = 5.1, p = 0.03, R^2 = 0.15$; Fig. 4), suggesting that animals with higher body condition values have lower BD. The slope and intercept of this relationship were substituted from Eqn. 6a to obtain the following equation:

$$(6b) \text{ BD} = 1000.71 - 278.12 * \text{BCI}$$

”

(16) Referee 1 – Comment (Line 401-402) – Suggest specifying this is between species as think unclear until next sentence.

Response to comment 16: We are grateful for the referee’s recommendation. We changed the text to detail that there is a significant difference between species.

Line 421-423: “A location-dispersion GLMM revealed significant differences *between species* in body measurements in terms of both their mean value (location; Fig. 5A), and variance around these mean values (dispersion; Fig. 5B).”

(17) Referee 1 – Comment (Line 407) – *Expand on what was the significant difference in measurements between mavic 3 drone and swimming condition.*

Response to comment 17: We value the referee’s observation to expand on this aspect of the results. We clarified that the significant difference associated with the Mavic 3 reflects a small (4 mm) but statistically detectable difference between the stationery and swimming conditions. We made this concept more explicit in the text and emphasized that, although statistically significant, the effect size was not biologically meaningful.

Line 425-428: “No significant differences *were attributed to the measurement method, with the exception for the Mavic 3, for which estimates obtained in the stationary condition were marginally (4 mm) but significantly higher than in the swimming condition (Table S4).*”

(18) Referee 1 – Comment (Line 411) – *May not be an issue but be good to expand briefly on this significant difference e.g. between mavic 3 and hand measurements or variation within mavic 3?*

Response to comment 18: We appreciate the referee pointing out the need for clarification. We clarified this point by moving the explanation of the significant difference between the stationery and swimming conditions for the Mavic 3 into the preceding paragraph, where the mean estimates are discussed. We expanded on the results and made them easier to follow. The section now reads as follow:

Line 425-433: “No significant differences *were attributed to the measurement method, with the exception for the Mavic 3, for which estimates obtained in the stationary condition were marginally (4 mm) but significantly higher than in the swimming condition (Table S4).*

Although statistically detectable, this difference is unlikely to be biologically meaningful (Fig. 5B; Table S4). Accuracy of UAV measurements was thus comparable to that of hand measurements.

Variation in body measurements was lower for UAV-based methods than for hand measurements, particularly when using a Mavic 3 (Fig. 5B). The UAV-based methods hence appear to have higher precision than hand measurements.”

(19) Referee 1 – Comment (Line 424) – *BCIBM was previously BCImass? Good to keep all these terms consistent for clarity.*

Response to comment 19: We thank the referee for noting this. We corrected the terms to maintain consistency.

Line 444: “showing that BCI_{BV} closely resembles the mass-based body condition index (BCI_{mass}).”

(20) Referee 1 – Comment (Line 519) – remove “particularly” as not needed. It is important.

Response to comment 20: We are grateful to the referee for the feedback. We implemented the suggestion, and the text now reads as follow:

Line 540-542: “This is important for marine mammal research, where photogrammetry offers a logistically feasible, cost-effective, and non-invasive method for assessing body condition.”

(21) Referee 1 – Comment (Line 524-526) – Need specific example where population dynamics comes in.

Response to comment 21: We value the referee’s input. We removed the reference to population dynamics in this section, as it was not directly relevant to the discussion of volumetric measurements. Population dynamics implications are now discussed later in the manuscript, within the section “Ecological implications.” The last sentence now reads:

Line 545-547: “This validation enables researchers to confidently use volume-based photogrammetry to investigate bioenergetics and physiology in free-ranging marine mammals.”

(22) Referee 1 – Comment (Line 532) – remove “remarkable”. Subjective term.

Response to comment 22: We appreciate the referee’s suggestion. We adjusted the text and removed the subjective term:

Line 553-556: “The consistency observed in mass estimates of both stationary and freely swimming subjects underscores the effectiveness of UAV photogrammetry as a reliable, non-invasive method for obtaining morphometric data in natural settings.”

(23) Referee 1 – Comment (Line 543-546) – *This is quite a strong statement. Suggest providing examples to help justify it*

Response to comment 23: We thank the referee for the recommendation. We revised the text to temper the original statement and to more directly align it with the empirical scope of our study.

Line 564-567: “By validating UAV photogrammetry for estimating body mass in odontocetes, our study provides empirical evidence that such estimates closely approximate the true body mass of living animals, supported by the strong agreement between photogrammetric estimates and manual mass measurements.”

(24) Referee 1 – Comment (Line 548-549) – *Suggest removing this sentence and going straight into the limitations. Since the title already indicates the focus, this is not necessary.*

Response to comment 24: We are grateful for the referee’s observation. We deleted the sentence and started discussing the limitations immediately after the title. The subsection reads as follow:

Line 568-570:

“4.4 Study limitations

Measurement errors due to water distortion or variations in body posture can affect photogrammetric measurements (Christiansen et al., 2018).”

(25) Referee 1 – Comment (Line 550-552) – *For a field setting it would be useful to know how many hours of video were taken and how many images were useful for measuring.*

Response to comment 25: We value the referee’s comment on this matter. We added more information on the number of recording hours and images collected during the swimming condition as this was simulating field settings.

Line 570-577: “We attempted to mitigate these factors by selecting images with minimal water spray, clear body contours, and straight postures, resulting in highly accurate estimates. For the free-swimming condition, we recorded an average eight minutes of video per individual and followed each animal for approximately six surfacing events. From these events, the single best image was selected for measurement. For field applications, where conditions are less controlled, we recommend recording at least ten surfacing events per individual, whenever possible, and then using the best single image for morphometric measurements.”

(26) Referee 1 – Comment (Line 557) – *Interesting to think about pregnancy and how that would change height:width ratio and body condition/mass. Something to add? Also was there any difference between males and females that could/should be investigated?*

Response to comment 26: The referee brings up an important point that we did not address. Photogrammetry studies of both toothed and baleen whales show that late-pregnant females have a visibly larger lateral body dimension, reflecting not only increased energetic stores but also the growing foetus, placenta, and associated fluids. This typically results in greater body width, especially at sites between 50 and 65% BL from the rostrum. Whether this increase in width is accompanied by a proportional increase in body height (dorso-ventral) remains unknown. Yet this information is essential for determining whether HW ratios at specific body sites change during pregnancy. No study to date has been able to quantify this, and we unfortunately did not have access to pregnant dolphins in our dataset. To acknowledge this limitation, we added the following text:

Line 598-604: “Aerial photogrammetry studies showed that late-pregnant females exhibit substantial expansion in width across the mid-body region in both toothed (30 – 60% of body length from the rostrum; Cheney et al., 2022) and baleen whales (50 – 65% of body length from the rostrum; Christiansen et al., 2022; van Aswegen et al., 2025), reflecting the physical space occupied by the foetus, placenta, and associated fluids. Whether this increase in width is matched by a proportional increase in dorso-ventral height remains unknown, as lateral images of late-pregnant individuals are currently unavailable.”

Response to comment 26: In response to the referee’s comment regarding potential sex difference in HW ratios, we re-ran a linear mixed-effect model to include sex as an additional fixed effect alongside body length, body condition, and species. This analysis revealed no detectable differences in HW ratios between males and females. To reflect this in the manuscript, we updated the Methods and Results to specify that sex was included as a fixed effect, added the corresponding statement in the discussion, and revised the Supplementary Table 2 to represent the updated model results:

Line 222-227: “We used a linear mixed-effects model using the package ‘glmmTMB’ (Brooks et al., 2017) to test whether the HW ratio was affected by species, body length (BL, adjusted by subtracting the species-specific mean from each individual according to species to address multicollinearity), body condition (expressed as the relative body widths W at 40% of body length measured from the rostrum), or sex with measurement site and individual ID included as random effects.”

Line 367-368: “The HW ratios were unaffected by BL, BCI (measured as W at 40% BL) and did not differ between species or sexes (Table S2).”

Line 605-608: “For non-pregnant adults, we found no difference in HW ratios between males and females (Table S2), suggesting similar cross-sectional body shape. This has important practical implications, as dolphins cannot be sexed reliably from aerial photographs, making sex-specific HW ratios difficult to apply in the field.”

Line 1080-1081: “

Effect Type	Term	Estimate	Std. Error	t value
Fixed	(Intercept)	2.324	0.562	4.133
	BL (centered)	-0.200	0.237	-0.842
	W at 40%BL	-3.425	3.304	-1.037
	Species (T.truncatus)	0.038	0.050	0.763
	Sex	-0.036	0.056	-0.645

”

Response to comment 26: The section “Study limitations” now reads as follow:

Line 569-608: “Measurement errors due to water distortion or variations in body posture can affect *photogrammetric measurements* (Christiansen et al., 2018). We attempted to mitigate these factors by selecting images with minimal *water* spray, clear body contours, and straight postures, resulting in highly accurate estimates. *For the free-swimming condition alone, we recorded on average eight minutes of video per individual and obtained approximately six pictures per animal, for which the single best image was selected for measurement. For field applications, where conditions are less controlled, we recommend recording at least ten pictures per individual whenever possible. The close agreement between UAV-derived and manual measurements indicates that distortion was not a major source of error in our dataset.*

In addition, because all measurements were collected at the same time of year, we were unable to capture seasonal changes in body condition (range: -0.186 to 0.267). Repeated measurements across different seasons could provide a more complete picture of the morphological range that dolphins naturally exhibit over the course of the year, as documented in other populations, such as Tamanend’s bottlenose dolphins (T. erebennus; Perkins-Taylor et al. 2024) and common bottlenose dolphins (Adamczak et al., 2021).

Our height:width (HW) ratios were derived from a dataset comprising mostly of adults and a few larger juveniles (n = 4; body length range: 1.96 – 2.99 m). The limited access to individuals from different age classes may not fully represent the ontogenetic variation in body shape, as allometric relationships describing how body proportions scale with length often shift during growth (Christiansen et al., 2019, 2020b). For instance, differences in HW ratios have been reported between calves and adults in right and humpback whales (Christiansen et al., 2019; Christiansen, Sprogis et al., 2020). Similar patterns may exist in bottlenose dolphins. Mallette et al. (2016) showed clear ontogenetic changes in body shape

and composition cross age classes in bottlenose dolphins, while Kurihara & Oda (2009) demonstrated allometric changes in skull morphology. Incorporating younger and smaller individuals in future photogrammetric studies will *therefore be essential for capturing the full range of morphometric development in bottlenose dolphins.*

Aerial photogrammetry studies showed that late-pregnant females exhibit substantial expansion in width across the mid-body region in both toothed (30 – 60% of body length from the rostrum; Cheney et al., 2022) and baleen whales (50 – 65% of body length from the rostrum; Christiansen et al., 2022; van Aswegen et al., 2025), reflecting the physical space occupied by the foetus, placenta, and associated fluids. Whether this increase in width is matched by a proportional increase in dorso-ventral height remains unknown, as lateral images of late-pregnant individuals are currently unavailable.

For non-pregnant adults, we found no difference in HW ratio between males and females (Table S2), suggesting similar cross-sectional body shape. This has important practical implications, as dolphins cannot be sexed reliably from aerial photographs, making sex-specific HW ratios difficult to apply in the field.”

(27) Referee 1 – Comment (Line 560) – *Were you able to see any different between the juveniles and adults? Even if not significant with small sample size.*

Response to comment 27: We thank the referee for commenting on this aspect. We agree that comparing HW ratios between age classes would be valuable. However, we chose not to conduct this analysis due to the very small number of juveniles ($n = 4$). With such limited representation, the results would be highly sensitive to influential data points, making any apparent differences difficult to interpret and potentially driven by sample size alone. As we cannot draw reliable conclusions at this stage, we prefer not to include analyses that our dataset cannot statistically support. We have instead clarified in the manuscript that our sample consists predominantly of adults and specified the low number of juveniles:

*Line 586-597: “Our height:width (HW) ratios were derived from a dataset comprising mostly of adults and a few larger juveniles ($n = 4$; body length range: 1.96 – 2.99 m). The limited access to individuals from different age classes may not fully represent the ontogenetic variation in body shape, as allometric relationships describing how body proportions scale with length often shift during growth (Christiansen et al., 2019, 2020b). For instance, significant differences in HW ratios have been reported between calves and adults in right and humpback whales (Christiansen et al., 2019; Christiansen, Sprogis et al., 2020). Similar patterns may exist in bottlenose dolphins. Mallette et al. (2016) showed clear ontogenetic changes in body shape and composition cross age classes in bottlenose dolphins, while Kurihara & Oda (2009) demonstrated allometric changes in skull morphology. Incorporating younger and smaller individuals in future photogrammetric studies will *therefore be essential for capturing the full range of morphometric development in bottlenose dolphins.*”*

(28) Referee 1 – Comment (Line 574-577) – Repeated from introduction.

Response to comment 28: We are grateful to the referee for pointing out this repetition. We revised the text by removing the repeated information and refocusing the paragraph on the broader implications of the validated method and its applications in demographic monitoring.

Line 610-616: “The validation of this method carries broad implications across ecological and evolutionary research. UAV photogrammetry is increasingly being used to obtain morphometric information from free-ranging cetaceans, enabling the assessment of age structure, demographic patterns, and population trajectories in the wild. For example, it has been applied to monitor a critically endangered bottlenose dolphin population in the Gulf of Ambracia, Greece, where precise body length measurements informed age distribution and provided insights into population stability (Vivier et al., 2024).”

(29) Referee 1 – Comment (Line 572-587) – Suggest condensing this paragraph, as the focus of the study is primarily on volume, mass, and condition, rather than body length.

Response to comment 29: We value the referee’s feedback. We condensed the text by discussing only one example focused on body length and demographic structure. The paragraph now reads as follow:

Line 610-620: “The validation of this method carries broad implications across ecological and evolutionary research. UAV photogrammetry is increasingly being used to obtain morphometric information from free-ranging cetaceans, enabling the assessment of age structure, demographic patterns, and population trajectories in the wild. For example, it has been applied to monitor a critically endangered bottlenose dolphin population in the Gulf of Ambracia, Greece, where precise body length measurements informed age distribution and provided insights into population stability (Vivier et al., 2024). Healthy populations typically show a balanced representation of calves, juveniles, and adults (Gamelon et al., 2016), while deviations from this distribution can signal population growth or decline (Coulson et al., 2005; Jones et al., 2018; Jackson et al., 2020). Monitoring shifts in body size over time can therefore provide early warning signals of population-level stressors.”

(30) Referee 1 – Comment (Line 588-601) – As above, important points, but suggest could be reduced as although the results from this study pave the way for future studies on maternal condition, this study didn’t investigate that.

Response to comment 30: We appreciate the referee for highlighting this. We condensed the text by briefly discussing the effect of body condition on female reproduction and how this relationship is documented in different taxa.

Line 621-628: “Beyond structural traits, body condition provides important information on the energetic status of individuals and their ability to meet the demands of growth, reproduction, and survival (Kleiber, 1947; Anderson-Teixeira et al., 2009). In particular, maternal condition, has strong implications for reproduction, *as* females in good condition tend to produce larger, more viable offspring (Kovacs & Lavigne, 1986; Tauson, 1993; Christiansen et al., 2018). Similar *relationships* between maternal condition and offspring survival *have been documented* in mysticetes (Christiansen et al., 2014, 2018), pinnipeds (Kovacs & Lavigne, 1986; McDonald et al., 2008), ungulates (Parker et al., 2009), and birds (Loretsen, 1996).”

(31) Referee 1 – Comment (Line 607-610) – Fits in more clearly to the next paragraph.

Response to comment 31: We thank the referee for helping to improve the text. We included this last sentence in the following paragraph. The new subsection starts now as such:

Line 636-640: “

4.6 Management and implications

At the population level, body condition also reflects nutritional health and demographic trends, making it a critical tool for wildlife monitoring and conservation (Stevenson & Woods, 2006; Rolland et al., 2016; Christiansen et al., 2020a). Marine ecosystems are rapidly changing under the influence of climate and other anthropogenic pressures (Crain et al., 2008; Hoegh-Guldberg & Bruno, 2010; Doney et al., 2012; Hawkins et al., 2017).”

Dear Mr Ciciarella,

Re: JP-TFP-2025-290419R1 "Validation of aerial photogrammetry methods to measure body size, condition, and mass in small cetaceans" by Riccardo Ciciarella, Erik P. Willems, Benjamin Markham, Manuela R. Bizzozzero, Wayne Phillips, Simon J. Allen, Michael Krützen, and Fredrik Christiansen

We are pleased to tell you that your paper has been accepted for publication in The Journal of Physiology.

Authors should note that it is too late at this point to offer corrections prior to proofing. Major corrections at proof stage, such as changes to figures, will be referred to the Editors for approval before they can be incorporated. Only minor changes, such as to style and consistency, should be made at proof stage. Changes that need to be made after proof stage will usually require a formal correction notice.

All queries at proof stage should be sent to: TJP@wiley.com

If you would like to receive our 'Research Roundup', a monthly newsletter highlighting the cutting-edge research published in The Physiological Society's family of journals (The Journal of Physiology, Experimental Physiology and Physiological Reports), please click this link, fill in your name and email address and select 'Research Roundup':
<https://www.physoc.org/journals-and-media/membernews/>

Yours sincerely,

Eleonora Grandi
Senior Editor
The Journal of Physiology

P.S. - You can help your research get the attention it deserves! Check out Wiley's free Promotion Guide for best-practice recommendations for promoting your work at www.wileyauthors.com/eoo/guide. You can learn more about Wiley Editing Services which offers professional video, design, and writing services to create shareable video abstracts, infographics, conference posters, lay summaries, and research news stories for your research at www.wileyauthors.com/eoo/promotion.

• **IMPORTANT NOTICE ABOUT OPEN ACCESS:** To assist authors whose funding agencies mandate immediate public access to published research findings, The Journal of Physiology allows authors to pay an Open Access (OA) fee to have their papers made freely available immediately on publication.

You can check if your funder or institution has a Wiley Open Access Account here: <https://authorservices.wiley.com/author-resources/Journal-Authors/licensing-and-open-access/open-access/author-compliance-tool.html>

EDITOR COMMENTS

Reviewing Editor:

Comments to the Author:

Thank you for revising the paper in line with the reviewers comments.

Senior Editor:

Comments to the Author:

Thank you for addressing the reviewers' comments.

REFEREE COMMENTS:

Referee #1:

The authors have carefully responded and revised the manuscript addressing all the comments raised in my previous review. I believe the changes have improved the clarity of the manuscript. I have no further suggestions and believe it is suitable for publication.

Referee #2:

I congratulate the authors on their thorough review of the manuscript.